# Persistent Backdoor Attacks in Class-Incremental Learning via Structural Invariant Anchoring

**Junhuang Huang** [* 1] **Linshan Hou** [* 2] **Jianting Ning** [1] **Yanjun Zhang** [3] **Zhongyun Hua** [2] **Leo Yu Zhang** [3]

## Abstract

Continual learning (CL) involves continual parameter updates, posing a significant challenge to backdoor persistence. In this paper, we reveal that the most advanced existing attack relies on an implicit assumption that task-critical neurons remain stable across task learning; however, this assumption does not hold in class-incremental learning (CIL). This exposes a critical research gap: backdoor persistence in CIL remains an open question. Inspired by functional stability, we discover that CIL models preserve task knowledge in shallow, structurally invariant subspaces. Motivated by these findings, we propose PBTO, the first persistent and targeted backdoor attack in CIL. PBTO trains a surrogate model on proxy tasks to obtain a parameter trajectory. It then optimizes a universal trigger that ensures misclassification to the target label across all model states and anchors trigger embeddings in shallow layers. Experimental results verify that PBTO maintains a high final attack success rate (ASR) across all benchmarks, while representative baselines degrade substantially after sequential learning. Code is available at PBTO.

## 1. Introduction

Akin to classical machine learning (ML), Continual Learning (CL) typically relies on external data for model training. This fundamental dependence on untrusted data exposes CL to backdoor attacks (Gu et al., 2017; Guo et al., 2025; Jiang et al., 2022). Adversaries inject a small set of poisoned samples into the training dataset. The poisoned samples contain designated triggers and are labeled with an attacker-specified target class, inducing a hidden backdoor association between the triggers and the target label during training. At inference, poisoned inputs stamped with the trigger are classified as the target class, whereas clean inputs remain unaffected. However, directly applying existing backdoor attacks to CL causes significant attack-performance degradation (Guo et al., 2025). This is primarily attributed to the dynamic nature of CL, where continuous parameter updates progressively eliminate the established backdoor associations. As a result, backdoor persistence emerges as a fundamental challenge in CL.

Existing backdoor attacks tailored to CL include three strategies, each attempting to address backdoor persistence but exhibiting limitations under Class-Incremental Learning (CIL) settings. Specifically, (1) continuous poisoning-based attacks (Gao & Liu, 2025) assume continuous injection of poisoned samples throughout CIL training, which is difficult to achieve in realistic deployment settings. (2) Task-Incremental Learning (TIL)-oriented backdoor attacks (Guo et al., 2025) rely on task-specific neurons to preserve the backdoor. However, CIL operates within a shared feature space and continuously repurposes existing neurons to accommodate new knowledge, inevitably undermining backdoor associations established under TIL assumptions. (3) CIL-oriented latent backdoor attacks (Jiang et al., 2022) uniformly suppress old-class logits for triggered samples, steering predictions away from the correct class rather than toward an attacker-specified target. Thus, they remain untargeted and cannot address targeted backdoor persistence in CIL. Consequently, a critical gap remains: no existing work simultaneously achieves one-time poisoning, persistence, and attacker-specified target control in CIL.

In this paper, we investigate the limitations of LTB (Guo et al., 2025), which, to the best of our knowledge, is the state-of-the-art backdoor attack in CL. We reveal that LTB's persistence relies on an assumption: task-critical neurons remain invariant across subsequent learning. However, we demonstrate that this assumption does not hold in CIL. Unlike TIL with task-specific architectures, CIL operates under a unified architecture with shared parameters continuously repurposed to adapt to new tasks. Interestingly, CIL models

---

*Equal contribution [1]Zhejiang Key Laboratory of Digital Fashion and Data Governance, Zhejiang Sci-Tech University, Hangzhou 310018, China [2]School of Computer Science and Technology, Harbin Institute of Technology, Shenzhen, China [3]Griffith University, Australia. Correspondence to: Jianting Ning <jtning88@gmail.com>.

*Proceedings of the 43rd International Conference on Machine Learning*, Seoul, South Korea. PMLR 306, 2026. Copyright 2026 by the author(s).

maintain robust performance on previous tasks even under severe neuron-level instability. This suggests that task knowledge is preserved through structural invariants at the subspace level rather than through fixed individual neurons. To verify this, we track the principal subspace of initial-task samples and measure its preservation via similarity analysis as new tasks are learned. We discover that shallow layers retain over 80% subspace similarity across tasks, while deeper layers exhibit significant drift. This reveals a critical insight for persistent backdoor design: anchor triggers to structurally invariant shallow subspaces.

Motivated by this insight, we propose Persistent Backdoor Trigger Optimization (PBTO), the first persistent and targeted backdoor attack for CIL. PBTO first simulates CIL parameter evolution by sequentially training a surrogate model on proxy tasks, constructing a model-state trajectory that mirrors real-world continual learning dynamics without accessing the victim model. Based on this trajectory, PBTO introduces a dual-objective optimization strategy to design a universal trigger that: (1) misclassifies clean samples to the attacker-specified target label across all surrogate model states, while (2) anchors trigger embeddings to a truly invariant structural subspace rather than trajectory-specific artifacts. PBTO achieves this by jointly minimizing a trajectory-averaged cross-entropy loss and a Gram-matrix regularization term that aligns trigger representations with target-class samples in shallow layers. To account for backdoor-induced gradient interference, PBTO further employs an iterative refinement strategy that alternates between simulating the poisoned trajectory and re-optimizing the trigger.

In summary, our contributions are four-fold. (1) We reveal a critical research gap: no existing work simultaneously achieves persistent and targeted backdoor attacks in CIL, leaving CIL robustness against such threats largely unexplored. (2) We discover that CIL models preserve task knowledge through structurally invariant subspaces in shallow layers. (3) Based on our findings, we propose PBTO, the first persistent and targeted backdoor attack for CIL systems, which anchors triggers to invariant structural subspaces via trajectory simulation and dual-objective optimization. (4) Experimental results indicate that PBTO achieves final ASRs of 97.1%, 86.5%, and 83.8% on CIFAR-10, CIFAR-100, and Tiny-ImageNet, respectively.

## 2. Related Work

### 2.1. Continual Learning

Continual learning trains a model on a stream of tasks while preserving knowledge from earlier stages (Parisi et al., 2019). Depending on what information is available at inference, CL is commonly studied under three settings (van de Ven & Tolias, 2019). In TIL, the task identity, which speci-

fies the task currently being performed by a CL model, is known at test time. Predictions can therefore be restricted to the corresponding task, often using task-specific components such as separate classifier heads. Domain-Incremental Learning (DIL) removes this task identity but keeps the label space shared across tasks, focusing on distribution shifts across domains. Class-Incremental Learning (CIL) is more challenging: new classes arrive over time, the task identity is unavailable at inference, and the model must predict over the unified label space of all classes learned so far.

This paper focuses on CIL. Although we use tasks to index the training stream, this index is not provided to the model during inference; all classes share a single feature extractor and classifier (Rebuffi et al., 2017; Buzzega et al., 2020). As new classes are incorporated, the same parameters are repeatedly updated and repurposed, making CIL particularly hostile to backdoor persistence: a trigger-target association learned at one stage can be overwritten by later learning. Existing CIL methods mitigate catastrophic forgetting mainly through replay-based methods (Rebuffi et al., 2017; Buzzega et al., 2020), regularization-based methods (Kirkpatrick et al., 2017; Li & Hoiem, 2017), and prompt-based methods (Wang et al., 2022a;b).

### 2.2. Backdoor Attacks

**Backdoor Attacks in Classical Learning.** In a backdoor attack, adversaries inject poisoned samples that contain designated triggers and are labeled as an attacker-specified target class into the training set. The trained model learns a hidden trigger-target association. At inference, triggered inputs are misclassified to the target class, while clean inputs remain unaffected. Early studies like BadNets (Gu et al., 2017) utilized visible patch-based triggers (Gu et al., 2017). To enhance stealthiness, subsequent research introduced invisible triggers via additive perturbations (Chen et al., 2017), image warping (Nguyen & Tran, 2021), or frequency-domain injection (Zeng et al., 2021). Based on the adversary's capabilities, attacks are generally categorized into poison-only, training-controlled, and model-modified settings (Li et al., 2022). In this paper, we focus on the poison-only setting, where the attacker can only modify a subset of training data.

**Backdoor Attacks in Continual Learning.** The security of CL against backdoor attacks remains largely underexplored. Existing studies either require continuous injection of poisoned samples throughout training (Gao & Liu, 2025), limiting practicality under realistic one-time poisoning constraints, or anchor triggers to architecture-specific components (Jiang et al., 2022) and task-specific neurons (Guo et al., 2025). These approaches either degrade under CIL's shared-parameter updates or, in the case of CIL-oriented latent backdoors, redirect triggered samples to an unspecified new class rather than an attacker-specified target. Con-

sequently, no existing work simultaneously achieves both persistence and target specification under realistic one-time poisoning constraints in CIL.

## 2.3. Backdoor Defenses

Backdoor defenses operate at two levels: (1) model-level defenses, which remove backdoors from trained models via neuron pruning (Liu et al., 2018) or knowledge distillation (Li et al., 2021b); and (2) input-level defenses that detect or neutralize triggered samples, including trigger inversion methods (Wang et al., 2019; Xu et al., 2024) and feature suppression techniques (Chen et al., 2025). These defenses generally operate under classical ML assumptions. We demonstrate that PBTO evades these defenses by coupling triggers with structural representations that overlap with benign features.

## 3. Revisiting Backdoor Persistence in CIL

### 3.1. Threat Model and Problem Formulation

**Threat Model.** In this paper, we consider the poison-only backdoor attack in class-incremental learning, which imposes minimal assumptions on the adversary. Specifically, we assume that the adversary can only poison the training data of one selected task by injecting poisoned samples, without access to the model architecture, parameters, gradients, or the specific incremental learning algorithm. Moreover, the adversary has no knowledge of the training distribution of the subsequent tasks to be learned after poisoning. Consistent with prior work, we assume the adversary has access to a small amount of publicly available data drawn from the same distribution as the attacker-specified target class.

**Attacker Goals.** In CIL, continuous parameter updates during incremental learning can erase the implanted backdoor. Thus, the adversary pursues three objectives. (1) Effectiveness: poisoned inputs are consistently misclassified to the attacker-specified target label. (2) Persistence: the backdoor remains effective throughout the continual learning process. Once the backdoor is injected during the learning of a target task, it should persist across all subsequent tasks. Despite continuous parameter updates induced by learning new incremental tasks, the model is expected to consistently misclassify poisoned samples to the target label at every stage of the continual learning process. (3) Transferability: the attack is effective across various model architectures.

**Poison-only Backdoor Attacks in CIL.** In CIL, a model is continuously trained to adapt to new tasks. Formally, for a sequence of tasks $\mathcal{T} = \{T_1, T_2, \dots, T_N\}$, the CIL objective is to achieve high accuracy on the current task $T_k$ (where $k \in \{1, 2, \dots, N\}$) and all previously learned tasks

$\{T_1, \dots, T_{k-1}\}$. Each task $T_k$, where $k \in \{1, 2, \dots, N\}$, provides a dataset $\mathcal{D}_k = \{(\boldsymbol{x}_i^{(k)}, y_i^{(k)})\}_{i=1}^{|\mathcal{D}_k|}$. The label space $\mathcal{Y}_k$ of task $T_k$ is disjoint from all other tasks, *i.e.*, $\mathcal{Y}_i \cap \mathcal{Y}_j = \emptyset$ for $i \neq j$. Starting from randomly initialized parameters $\boldsymbol{\theta}_0$, at each step $k$, the parameters $\boldsymbol{\theta}_{k-1}$ are updated on the dataset $\mathcal{D}_k$, producing a parameter trajectory $\boldsymbol{\theta}_0 \xrightarrow{\mathcal{D}_1} \boldsymbol{\theta}_1 \xrightarrow{\mathcal{D}_2} \boldsymbol{\theta}_2 \xrightarrow{\mathcal{D}_3} \cdots \xrightarrow{\mathcal{D}_N} \boldsymbol{\theta}_N$. The final model $\mathbf{F}_{\boldsymbol{\theta}_N}$ (also denoted as $\mathbf{F}_{\boldsymbol{\theta}}$) performs inference over the cumulative label space $\mathcal{Y} = \bigcup_{j=1}^{N} \mathcal{Y}_j$.

In the poison-only backdoor attack, attackers can only inject poisoned samples into the training set of a specific target task. Formally, for a selected task $T_k$ with its training set $\mathcal{D}_k$, the attacker first selects a subset of clean samples $\mathcal{D}_s \subset \mathcal{D}_k$. Each clean sample $(\boldsymbol{x}, y) \in \mathcal{D}_s$ is then transformed into a poisoned sample $(\hat{\boldsymbol{x}}, y_t)$, $\hat{\boldsymbol{x}} := \mathcal{G}(\boldsymbol{x}, \boldsymbol{\delta}) = \boldsymbol{x} \oplus \boldsymbol{\delta}$, where $\boldsymbol{\delta}$ is the trigger, and $y_t \in \mathcal{Y}_k$ is the attacker-specified target label. Then, the poisoned sample subset $\mathcal{D}_p$ is constructed: $\mathcal{D}_p = \{(\mathcal{G}(\boldsymbol{x}, \boldsymbol{\delta}), y_t) \mid (\boldsymbol{x}, y) \in \mathcal{D}_s\}$. The final poisoned training set for task $T_k$ is constructed by replacing $\mathcal{D}_s$ with the poisoned samples $\mathcal{D}_p$, yielding the poisoned training set $\hat{\mathcal{D}}_k = (\mathcal{D}_k \setminus \mathcal{D}_s) \cup \mathcal{D}_p$. After training on the poisoned dataset $\hat{\mathcal{D}}_k$, the model persistently behaves as:

$$\hat{\mathbf{F}}_{\boldsymbol{\theta}_j}(\boldsymbol{x}) = y, \quad \hat{\mathbf{F}}_{\boldsymbol{\theta}_j}(\hat{\boldsymbol{x}}) = y_t, \quad \forall j \in \{k, \dots, N\}, \quad (1)$$

where clean inputs $\boldsymbol{x}$ are correctly classified, and poisoned inputs $\hat{\boldsymbol{x}}$ are consistently misclassified to the target label $y_t$ even as the model updates parameters on subsequent tasks.

### 3.2. Rethinking the Neuron-Level Stability

To the best of our knowledge, LTB (Guo et al., 2025) currently represents the most advanced persistent backdoor attack for Task-Incremental Learning (TIL). It assumes that task-critical neurons, characterized by high Fisher information, remain isolated and their parameter values stay unchanged during subsequent training. This assumption indeed holds in TIL, where task identities are explicitly provided, allowing task-specific architectures (*e.g.*, separate classifier heads) to preserve isolated parameter subspaces for each task. In contrast, in CIL, task identities are unavailable, and all tasks share a unified model architecture. This architectural constraint fundamentally breaks the neuron isolation assumption in LTB: as new tasks are learned, the shared network continuously repurposes existing neurons to accommodate new knowledge, inevitably overwriting previously critical parameters.

To validate the instability of task-critical neurons in CIL, we conduct an analysis by tracking parameter dynamics throughout continual learning. Specifically, we train a ResNet-18 model on CIFAR-10 in a five-task CIL setting, where each task introduces two new classes sequentially. We identify two sets of neurons: (1) neurons that are critical

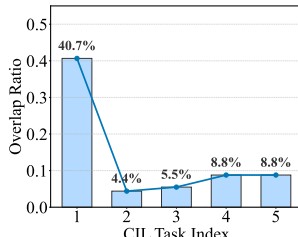
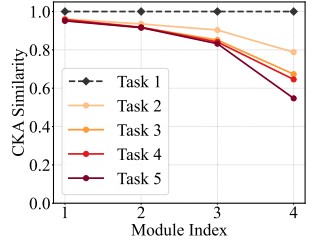

*Figure 1.* Stability of task-critical neurons in CIL, quantified by the overlap ratio between critical neurons and stable neurons across five tasks.

*Figure 2.* Stability of module-wise feature representations, measured by CKA similarity between Task 1 representations and subsequent tasks.

*Table 1.* Complementary subspace stability metrics on ResNet-18/CIFAR-100 after ten CIL tasks. Smaller Grassmann distance and larger variance retention indicate more stable subspaces.

| Module | Grassmann Distance ↓ | Variance Retention ↑ |
|--------|---------------------|---------------------|
| Module 1 | 0.08 | 0.92 |
| Module 2 | 0.12 | 0.88 |
| Module 3 | 0.38 | 0.54 |
| Module 4 | 0.45 | 0.47 |

for Task 1, determined by high Fisher information values, and (2) neurons whose parameters change minimally across all five tasks. We then measure the overlap ratio between these two sets to assess whether task-critical neurons remain stable. As shown in Figure 1, this overlap ratio drops to below 10%, demonstrating a severe divergence between task-criticality and parameter stability under CIL's continuous feature repurposing pressure. This finding confirms that the neuron-isolation assumption of LTB fails to hold in CIL.

### 3.3. From Neuron Isolation to Subspace Anchoring

The analysis in Sec. 3.2 reveals severe instability of individual neurons; however, CIL models maintain robust performance on previously learned tasks, which is CIL's fundamental objective. This observation suggests that task-relevant information may be preserved not solely through fixed neuron values, but potentially via invariant geometric structures in the feature space. This motivates us to shift perspective: instead of tracking individual neurons, we investigate subspace-level invariance across CIL.

To validate the existence of structural invariance, we adopt a natural strategy: tracking the feature subspace across early tasks to observe whether it remains stable after learning new tasks. If an invariant subspace exists, the principal structure captured from the initial task should remain intact during the subsequent tasks' learning. We operationalize this by first extracting the dominant structural representation from Task 1 at a specific layer $\ell$, forming feature matrix $\mathbf{\Phi}_\ell^{(1)}$. To establish a reference subspace that captures the most principal directions, we apply Singular Value Decomposition (SVD) to $\mathbf{\Phi}_\ell^{(1)} = \mathbf{U}\mathbf{\Sigma}\mathbf{V}^\top$. The first $r$ left singular vectors $[\mathbf{u}_1, \ldots, \mathbf{u}_r]$ of $\mathbf{U}_r$, corresponding to the largest singular values, define our $r$-dimensional principal reference subspace. As training progresses through Tasks 2, 3, …, $N$, we re-extract features for Task 1 samples using the updated model, obtaining $\mathbf{\Phi}_\ell^{(1|t)}$. We measure structural preservation from three complementary views. CKA (Kornblith et al., 2019) quantifies representation-space similarity while remaining invariant to unitary transformations. Grassmann

distance directly measures the geometric distance between principal subspaces. Projection-based variance retention measures how much later-stage representations can still be explained by the initial-task subspace.

Using the same five-task CIL setting as in Sec. 3.2, we analyze representational stability by extracting features from 1,000 Task 1 test samples after each of the four residual blocks, denoted as Modules 1–4 according to network depth, at different training stages. Figure 2 shows that this stability is not uniform across depth: shallow layers maintain high CKA similarity across tasks, while the final residual block exhibits significant representational drift.

This observation is further validated by the Grassmann distance and projection-based variance retention metrics after ten CIL tasks, as summarized in Table 1. Specifically, Module 1 exhibits a small Grassmann distance of 0.08 and retains 92% of the initial-task variance, indicating strong preservation of the original subspace structure. In contrast, Module 4 reaches a Grassmann distance of 0.45 while retaining only 47% variance, suggesting severe subspace deviation in the model's deeper representations. These consistent observations across both similarity- and geometry-based metrics indicate that shallow layers preserve a relatively stable representational subspace throughout continual learning, whereas deeper layers become progressively task-adaptive. Additional temporal and cross-architecture analyses on VGG-16 and ViT-Small/16 are provided in Appendix B.

**Insights for Persistent Attack Design.** The findings reveal a critical vulnerability in CIL: the structurally invariant subspace in shallow layers provides a stable anchor point for backdoor attacks. Since low-level representations remain stable throughout CIL, backdoor triggers embedded within this invariant subspace can persist across tasks. This observation motivates our core attack strategy: anchor the backdoor within the invariant subspace of early layers, exploiting the structural stability that enables continual learning to achieve persistent backdoor attacks throughout CIL.

## 4. Persistent Backdoor Trigger Optimization

### 4.1. Overview

Building on the insights in Sec. 3.3, we propose Persistent Backdoor Trigger Optimization (PBTO), the first persistent

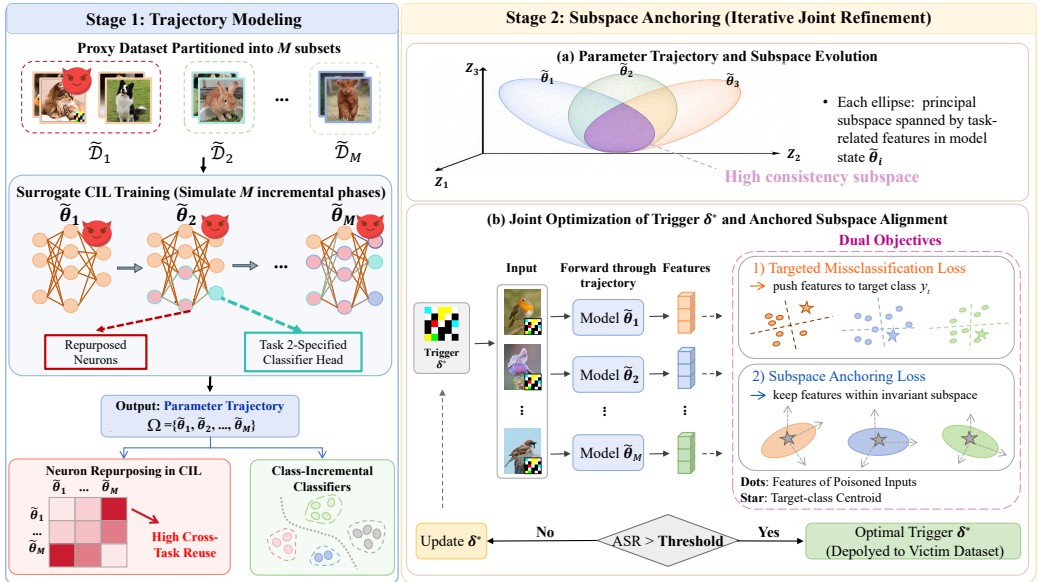

*Figure 3.* The main pipeline of PBTO. **Stage 1. Trajectory Modeling**: PBTO trains a surrogate model on proxy tasks to simulate CIL parameter evolution, obtaining a trajectory of model states. **Stage 2. Subspace Anchoring**: PBTO optimizes a universal trigger through dual-objective optimization and then iteratively refines the trigger by alternating between trajectory simulation and trigger optimization.

backdoor attack designed for the CIL setting. Specifically, PBTO mainly consists of two stages as shown in Fig. 3: (1) Trajectory Modeling, where a surrogate model is trained on proxy tasks to simulate the parameter evolution induced by CIL, thereby constructing an approximate model-state trajectory during the CIL process; and (2) Subspace Anchoring, which optimizes backdoor triggers to induce misclassification of clean inputs to the target label, while simultaneously anchoring the trigger representations in the invariant shallow subspace shared across different model states along the simulated parameter trajectory.

## 4.2. Trajectory Modeling

The continual parameter evolution in CIL challenges backdoor persistence, as triggers optimized on a single model state are easily disrupted by subsequent feature repurposing. As revealed in Sec. 3.3, shallow feature subspaces exhibit greater stability across CIL tasks. Ideally, a persistent trigger should anchor its representations within such invariant subspaces. However, in poison-only attacks, the victim's subsequent tasks are unavailable. This raises a key question: *how can triggers be anchored to stable subspaces without access to the CIL process?*

To address this challenge, we proactively simulate the parameter evolution process in CIL, thereby approximating the parameter-update trajectory. The key insight is that CIL models tend to reuse similar shallow structural subspaces for the same target class, even when subsequent tasks are constructed with different class partitions. This allows the attacker to approximate the victim's future update trajectory by constructing surrogate CIL sequences with proxy classes,

without knowing the victim's exact future classes or task partitions. Specifically, we assume the attacker has a local proxy dataset $\tilde{\mathcal{D}}$, independently collected from public data sources or generated images, and uses it to form surrogate CIL tasks. This assumption aligns with the threat model commonly adopted in existing backdoor attacks, where attackers possess auxiliary data resources. To emulate the continual learning dynamics, we partition the proxy dataset $\tilde{\mathcal{D}}$ into $M$ disjoint subsets:

$$\tilde{\mathcal{D}} = \bigcup_{m=1}^{M} \tilde{\mathcal{D}}_m, \quad \tilde{\mathcal{D}}_i \cap \tilde{\mathcal{D}}_j = \emptyset \text{ for } i \neq j. \quad (2)$$

We then sequentially train a surrogate model on these subsets to simulate $M$ incremental learning phases. This process collects a locally simulated proxy trajectory:

$$\Omega = \{\tilde{\boldsymbol{\theta}}_1, \ldots, \tilde{\boldsymbol{\theta}}_M\}. \quad (3)$$

## 4.3. Trigger Optimization via Subspace Anchoring

The trajectory modeling in Sec. 4.2 yields a sequence of surrogate model states $\Omega$ that approximates the victim's parameter evolution. Building on this trajectory, we design a trigger optimization strategy that simultaneously achieves two objectives: (1) trajectory-invariant optimization, which ensures the trigger successfully induces misclassification across all simulated model states; and (2) shallow subspace anchoring, which constrains the trigger to the stable shallow subspace. By jointly optimizing these two objectives, the trigger is continually refined and finally anchored to features that remain both effective and stable across all parameter states during the proxy CIL process, thereby achieving persistent backdoor behavior in real CIL scenarios.

**Trajectory-Invariant Optimization.** We first optimize a universal trigger $\boldsymbol{\delta}$ that maintains effectiveness across all model states in $\Omega$. For a specific task $T_k$, an attacker selects a subset $\mathcal{D}_s \subset \mathcal{D}_k$ of clean samples for poisoning. For each poisoned input $\hat{\boldsymbol{x}} = \boldsymbol{x} \oplus \boldsymbol{\delta}$ (where $\boldsymbol{x} \in \mathcal{D}_s$), the objective is to consistently redirect the prediction to target label $y_t$ across the entire trajectory. We formulate this as minimizing the averaged cross-entropy loss over all surrogate states:

$$\mathcal{L}_t(\boldsymbol{\delta}) = \frac{1}{|\Omega||\mathcal{D}_s|} \sum_{\tilde{\boldsymbol{\theta}}_m \in \Omega} \sum_{\boldsymbol{x} \in \mathcal{D}_s} \mathcal{L}_c\Big(\tilde{\mathbf{F}}_{\tilde{\boldsymbol{\theta}}_m}(\boldsymbol{x} \oplus \boldsymbol{\delta}), y_t\Big), \quad (4)$$

where $\tilde{\mathbf{F}}_{\tilde{\boldsymbol{\theta}}_m}$ denotes the $m$-th surrogate model and $\mathcal{L}_c$ is the cross-entropy loss function. By optimizing across the entire trajectory rather than a single model snapshot, the trigger is forced to exploit features that persist in CIL.

**Shallow Subspace Anchoring.** While $\mathcal{L}_t$ enforces backdoor effectiveness across the trajectory, it does not guarantee the trigger exploits the invariant shallow subspace. As a result, the optimization may converge to features stable across proxy models but not inherently invariant, which may fail on unseen tasks. To address this, we introduce a regularization term that explicitly anchors the trigger to stable shallow layers. Inspired by Independent Subspace Analysis (Hyvärinen & Hoyer, 2000), we employ Gram matrices (Gatys et al., 2016) to capture channel-wise correlations in shallow features, encoding feature statistics invariant to spatial configurations. Let $\phi_\ell(\cdot; \tilde{\boldsymbol{\theta}}_m)$ denote the feature extractor up to shallow layer $\ell$ of surrogate model $\tilde{\mathbf{F}}_{\tilde{\boldsymbol{\theta}}_m}$ parameterized by $\tilde{\boldsymbol{\theta}}_m$. We define the anchoring loss as:

$$\mathcal{L}_s(\boldsymbol{\delta}) = \frac{1}{|\Omega||\mathcal{D}_s|} \sum_{\Omega} \sum_{\boldsymbol{x} \in \mathcal{D}_s} \left\| \mathbf{G}_\ell^{(m)}(\boldsymbol{x} \oplus \boldsymbol{\delta}) - \mathbf{G}_\ell^{(m)}(\boldsymbol{r}) \right\|_F^2, \quad (5)$$

where $\mathbf{G}_\ell^{(m)}(\cdot) = \mathbf{G}(\phi_\ell(\cdot; \tilde{\boldsymbol{\theta}}_m))$ denotes the Gram matrix of shallow layer $\ell$, and $\boldsymbol{r}$ is a reference sample from the target class $y_t$ in the proxy dataset $\tilde{\mathcal{D}}$. Typically, we use ResNet-18 and extract features from the penultimate residual block, which our CKA analysis identified as exhibiting high cross-task stability. By minimizing this loss across the entire trajectory, the trigger is constrained to produce shallow feature correlations similar to genuine target-class samples, thereby anchoring to the stable subspace.

**Iterative Joint Refinement.** The above trigger optimization is performed on the backdoor-free surrogate models. However, in reality, once the attacker injects the backdoor at task $T_k$, all subsequent tasks $\{k+1, \ldots, N\}$ are learned on top of the already poisoned model. This creates a trajectory shift: the trigger is optimized on a clean trajectory, but the actual attack unfolds along a poisoned trajectory where backdoor-induced gradient interference alters subsequent learning dynamics. To address this mismatch, we propose an iterative refinement strategy that alternates between trigger

optimization and surrogate model training. The complete process is described as follows:

**Initialization ($t = 0$).** Train $M$ backdoor-free surrogate models sequentially on clean proxy datasets $\{\tilde{\mathcal{D}}_1, \ldots, \tilde{\mathcal{D}}_M\}$, yielding the initial clean trajectory $\Omega^{(0)} = \{\tilde{\boldsymbol{\theta}}_1^{(0)}, \ldots, \tilde{\boldsymbol{\theta}}_M^{(0)}\}$. Optimize the initial trigger with $\Omega^{(0)}$:

$$\boldsymbol{\delta}^{(0)} = \arg\min_{\boldsymbol{\delta}} \left[ \mathcal{L}_t(\boldsymbol{\delta}) + \lambda \mathcal{L}_s(\boldsymbol{\delta}) \right], \quad (6)$$

where $\lambda > 0$ is a hyperparameter balancing the two losses.

**Iterative Refinement ($t \geq 1$).** For each iteration $t$, perform two alternating steps:

*Step 1: Trajectory Simulation.* Using trigger $\boldsymbol{\delta}^{(t-1)}$, construct the poisoned dataset for task $T_k$:

$$\tilde{\mathcal{D}}_k^{(t)} = \{(\boldsymbol{x} \oplus \boldsymbol{\delta}^{(t-1)}, y_t) \mid \boldsymbol{x} \in \mathcal{D}_s\} \cup \{\mathcal{D}_k \setminus \mathcal{D}_s\}. \quad (7)$$

Simulate the CIL process: train sequentially on tasks $\{T_1, \ldots, T_{k-1}\}$ with clean data, task $T_k$ with poisoned data $\tilde{\mathcal{D}}_k^{(t)}$, and tasks $\{T_{k+1}, \ldots, T_M\}$ with clean data, yielding the poisoned trajectory:

$$\Omega^{(t)} = \{\tilde{\boldsymbol{\theta}}_1^{(t)}, \ldots, \tilde{\boldsymbol{\theta}}_M^{(t)}\}. \quad (8)$$

*Step 2: Trigger Re-optimization.* Fix trajectory $\Omega^{(t)}$ and update the trigger $\boldsymbol{\delta}^{(t)}$ same as in Eq. (6).

*Step 3: Convergence Check.* After obtaining $\boldsymbol{\delta}^{(t)}$, compute the attack success rate (ASR) across all proxy tasks:

$$\text{ASR}^{(t)} = \frac{1}{M \cdot |\mathcal{D}_s|} \sum_{m=1}^{M} \sum_{\boldsymbol{x} \in \mathcal{D}_s} \mathbb{I}\left[ \tilde{\mathbf{F}}_{\tilde{\boldsymbol{\theta}}_m^{(t)}}(\boldsymbol{x} \oplus \boldsymbol{\delta}^{(t)}) = y_t \right], \quad (9)$$

where $\mathbb{I}[\cdot]$ is the indicator function. The refinement stops when $\text{ASR}^{(t)} \geq \tau$, when the ASR improvement between two consecutive rounds is below $\epsilon$, or when the number of iterations reaches $T_{\max}$. Here, $\tau \in (0, 1]$ is the ASR threshold, $\epsilon > 0$ is the convergence tolerance, and $T_{\max}$ is the maximum number of iterations. By iteratively coupling trajectory simulation and trigger optimization, the final trigger $\boldsymbol{\delta}^*$ adapts to the poisoned training updates.

After obtaining the optimized trigger $\boldsymbol{\delta}^*$, the attacker constructs a poisoned training set for the selected task by injecting the trigger into a small subset of its clean samples and relabeling them as the attacker-specified target label $y_t$. When the victim model is trained on this poisoned dataset, the backdoor becomes embedded into the model's structural invariants. As a consequence, the malicious backdoor behavior remains effective throughout the continual learning process, posing a persistent threat to CIL security.

*Table 2.* ASR and BA after learning all $N$ CIL tasks. Results are averaged over 3 runs ($\pm$ Std. Dev.).

| Method | CIFAR-10 | | CIFAR-100 | | Tiny-ImageNet | |
|---|---|---|---|---|---|---|
| | ASR ($\uparrow$) | BA | ASR ($\uparrow$) | BA | ASR ($\uparrow$) | BA |
| BadNets | 12.4$\pm$1.5 | 82.3$\pm$0.4 | 4.5$\pm$0.5 | 68.5$\pm$0.7 | 2.1$\pm$0.3 | 58.4$\pm$0.9 |
| WaNet | 8.5$\pm$2.1 | 81.9$\pm$0.5 | 3.2$\pm$1.1 | 68.1$\pm$0.6 | 1.5$\pm$0.4 | 57.9$\pm$0.8 |
| DRUPE | 28.5$\pm$3.2 | 82.0$\pm$0.6 | 14.2$\pm$1.9 | 67.9$\pm$0.8 | 5.6$\pm$1.1 | 58.0$\pm$1.0 |
| LTB | 12.4$\pm$3.5 | 81.8$\pm$0.5 | 4.7$\pm$3.2 | 67.5$\pm$1.2 | 8.9$\pm$2.4 | 57.8$\pm$1.4 |
| **PBTO (Ours)** | **97.1**$\pm$0.4 | 81.6$\pm$0.3 | **86.5**$\pm$0.9 | 67.2$\pm$0.6 | **83.8**$\pm$1.1 | 57.5$\pm$1.7 |

## 5. Experiments

### 5.1. Experimental Setup

**Datasets and Benchmarks.** We evaluate PBTO on CIFAR-10, CIFAR-100, and Tiny-ImageNet under standard class-incremental protocols (Rebuffi et al., 2017), where classes are partitioned into $N$ disjoint tasks and learned sequentially. We set $N = 5$ for CIFAR-10 and $N = 10$ for all other datasets. Results are reported as mean $\pm$ standard deviation over 3 independent runs with distinct random seeds, covering variations in task ordering and initialization. We report attack success rate (ASR) and benign accuracy (BA).

**Victim Model & Protocol.** We adopt iCaRL (Rebuffi et al., 2017) with a ResNet-18 backbone and a memory budget of $\mathcal{M} = 2,000$ exemplars. The attacker poisons only Task 1, which creates the longest retention interval and tests whether the backdoor remains effective after subsequent tasks.

**Proxy Dataset.** PBTO constructs a proxy dataset $\tilde{\mathcal{D}}$ using only public or generated images, without accessing the victim's private training samples. We split $\tilde{\mathcal{D}}$ into $M = 5$ proxy tasks. Each task contains $K$ proxy classes, where $K$ is set to 5 by default. Each proxy class contains 5,000 object-centric images. We collect candidate images using class names or related queries and filter them manually or with a classifier. We use a pre-trained DDPM (Ho et al., 2020) to supplement any remaining data shortage. All images are resized to match the input resolution of CIFAR-10, CIFAR-100, or Tiny-ImageNet.

**Baselines.** We compare PBTO with three categories of backdoor baselines relevant to persistent attacks in CIL. (1) Classical backdoor attacks with different trigger designs, including BadNets (Gu et al., 2017), WaNet (Nguyen & Tran, 2021), and DRUPE (Tao et al., 2024). (2) Advanced backdoor attacks, including ISSBA (Li et al., 2021a), LIRA (Doan et al., 2021), and WaveAttack (Xia et al., 2024), reported in Appendix D. (3) LTB (Guo et al., 2025), the most advanced persistent backdoor attack proposed for continual learning under the TIL setting. To adapt LTB to CIL, we replace its multi-head architecture with a unified classifier.

**Implementation Details.** To ensure trigger stealthiness, we restrict the perturbation magnitude of the optimized trigger,

*Table 3.* Cross-architecture transferability of PBTO measured by ASR (%). Triggers are optimized on a ResNet-18 source surrogate and evaluated on unseen target architectures.

| Source | Target | Data | LTB | PBTO |
|---|---|---|---|---|
| ResNet-18 | VGG-16 | CIFAR-10 | 5.5% | **78.1%** |
| | DenseNet-121 | CIFAR-10 | 6.3% | **82.2%** |
| | MobileNetV2 | CIFAR-10 | 5.8% | **81.5%** |
| | ViT-Small/16 | CIFAR-100 | 4.1% | **68.7%** |

i.e., $\|\boldsymbol{\delta}\|_\infty \leq \epsilon$ with $\epsilon = 8/255$, and poison only $\rho = 0.05$ of Task-1 training samples. The trigger $\boldsymbol{\delta}$ is optimized via PGD under this $L_\infty$ constraint with $\lambda = 1.0$. All baselines are reproduced with their official configurations.

### 5.2. Main Results

To evaluate attack effectiveness, we report ASR and BA after the final task, where the backdoor faces the most challenging persistence setting. The backdoor is injected only once by poisoning the training set before Task 1. As shown in Table 2, PBTO achieves the highest ASR on all datasets while maintaining BA comparable to the baselines. In contrast, all baselines degrade sharply after sequential learning. The most advanced LTB attack obtains only 4.7% ASR on CIFAR-100 and 8.9% ASR on Tiny-ImageNet. Appendix F further shows task-wise results. PBTO decreases gradually from 98.0% after Task 1 to 86.5% after learning ten tasks on CIFAR-100, whereas all other attacks drop below 14.2

### 5.3. Transferability

**Cross Surrogate-Victim Architecture Transferability.** To evaluate cross-architecture transferability of PBTO, we optimize triggers on a ResNet-18 surrogate and evaluate them on unseen target architectures, including VGG-16, DenseNet-121, MobileNetV2, and ViT-Small/16.

Table 3 shows that triggers optimized on ResNet-18 still obtain 78.1%, 82.2%, and 81.5% ASR on VGG-16, DenseNet-121, and MobileNetV2, respectively. The CNN-to-ViT transfer is harder but still retains 68.7% ASR, indicating that the transfer is not restricted to a CNN family. Additional scalability results on the larger and higher-resolution VGGFace2 dataset (Cao et al., 2018) are reported in Appendix E, where PBTO reaches 85.9% ASR after ten tasks. These results

*Table 4.* ASR (%) on CIFAR-100 after ten CIL tasks under representative regularization- and replay-based CIL algorithms.

| CL Strategy | Algorithm | LTB (ASR) | PBTO (ASR) |
|---|---|---|---|
| Regularization | EWC | 8.2% | **88.1%** |
| Replay | iCaRL | 4.7% | **86.5%** |
| Replay (Strong) | DER++ | 5.4% | **85.8%** |

indicate that PBTO is not simply memorizing one source-model parameter trajectory.

**Cross Surrogate-Victim Training Protocol Transferability.** To verify PBTO's transferability under imperfect surrogate-victim knowledge, we evaluate whether a trigger optimized with a surrogate remains effective when the victim differs in training protocols including task order, data distribution, or training hyperparameters. Specifically, we test five random task orderings, cross-dataset surrogate-victim pairs between CIFAR-100 and Tiny-ImageNet, and mismatched victim learning rates or batch sizes, with full settings reported in Appendix G. PBTO remains stable across task-order changes, achieving an average ASR of $86.1 \pm 1.0\%$. Training-hyperparameter mismatch has only a minor effect, with ASR staying within 83.1–87.2%, whereas cross-dataset mismatch is more challenging but still retains 69.3–72.4% ASR. Overall, these results demonstrate that PBTO transfers across surrogate-victim task orders, data distributions, and training configurations, indicating that it does not rely on an exact surrogate-victim trajectory match.

## 5.4. Performance Across CL Paradigms.

Another practical concern is whether PBTO only works with the iCaRL training dynamics used in the main benchmark. Table 4 evaluates PBTO under representative regularization- and replay-based CIL strategies. Across EWC (Kirkpatrick et al., 2017), iCaRL (Rebuffi et al., 2017), and DER++ (Buzzega et al., 2020), PBTO remains above 85% ASR, whereas LTB stays below 9%. This result supports anchoring the trigger to structural representations that continue to be reused during later tasks, rather than to isolated critical neurons.

## 5.5. Resilience Against Defenses

To evaluate PBTO's resistance against existing defenses, we apply representative post-training defenses after the victim completes the CIL sequence. Specifically, we consider model-mitigation defenses, including Fine-Pruning (Liu et al., 2018) and Neural Attention Distillation (NAD) (Li et al., 2021b), and input purification or reconstruction defenses, including BTI-DBF (Xu et al., 2024) and RE-FINE (Chen et al., 2025); PBTO is injected in Task 1, and each defense is applied after the final CIL task.

As shown in Table 5, PBTO still retains high ASR under

*Table 5.* Resistance of PBTO against existing defenses. The attack is injected before Task 1, while defenses are applied after the final task.

| Defense Mechanism | ASR (%) |
|---|---|
| *No Defense* | 97.1±0.4 |
| Fine-Pruning | 88.4±1.2 |
| NAD | 82.1±1.8 |
| BTI-DBF | 89.5±0.9 |
| REFINE | 86.2±1.5 |

*Table 6.* Task-wise ASR(%) under different defenses.

| Defense | T1 | T2 | T3 | T4 | T5 |
|---|---|---|---|---|---|
| *No Defense* | 99.3 | 98.6 | 98.0 | 97.5 | 97.1 |
| Fine-Pruning | 93.8 | 92.1 | 90.6 | 89.4 | 88.4 |
| NAD | 89.5 | 87.3 | 85.2 | 83.5 | 82.1 |
| BTI-DBF | 94.6 | 93.2 | 91.8 | 90.5 | 89.5 |
| REFINE | 92.0 | 90.5 | 88.8 | 87.4 | 86.2 |

all defenses, with 88.4% after Fine-Pruning, 82.1% after NAD, 89.5% after BTI-DBF, and 86.2% after REFINE. Table 6 further shows that the defended ASR remains high across all five CIL stages, indicating that this behavior is not merely an endpoint effect. These results suggest that PBTO is resistant to existing post-training defenses because its backdoor association is coupled with stable structural representations rather than isolated trigger artifacts or easily removable inactive neurons.

## 5.6. Ablation Study

**Contribution of Key Components.** PBTO contains two core designs: surrogate trajectory simulation anticipates future parameter drift, while the representation-alignment constraint $\mathcal{L}_s$ encourages poisoned samples to follow stable structural representations. Table 7 decouples these two effects. Without either component, the attack reaches only 4.5% ASR after continual learning. Using either trajectory simulation or $\mathcal{L}_s$ alone improves ASR to 68.9% and 62.1%, respectively, while combining them reaches 86.5% ASR. This shows that both trajectory use and representation alignment are necessary for backdoor persistence.

*Table 7.* Component analysis on CIFAR-100 after ten CIL tasks. We decouple Surrogate Trajectory Simulation (Sim) and the representation-alignment constraint ($\mathcal{L}_s$).

| Sim (Trajectory) | $\mathcal{L}_s$ (Align) | ASR (%) | BA (%) |
|---|---|---|---|
| ✗ | ✗ | 4.5 | 68.8 |
| ✓ | ✗ | 68.9 | 67.5 |
| ✗ | ✓ | 62.1 | 66.9 |
| ✓ | ✓ | **86.5** | 67.2 |

**Simulation Trajectory Length.** The number of surrogate

phases $M$ controls how much of the future CIL trajectory is covered during trigger optimization. This ablation tests whether PBTO must simulate the entire future task sequence or can persist beyond a moderate surrogate horizon. Table 8 varies $M$ from 1 to 10. In our standard CIFAR-100 setting, the victim is evaluated after ten CIL tasks, whereas the default surrogate trajectory uses $M = 5$ phases; thus Tasks 6–10 already fall outside the simulated horizon. A single simulated state gives only 65.2% ASR, indicating that the trigger overfits to the immediate model snapshot. ASR improves as the trajectory covers more CIL phases and saturates at $M = 5$; increasing $M$ to 10 yields the same final ASR of 86.5%. Together with the task-wise ASR trend in Appendix F, this suggests that PBTO captures reusable structural patterns that remain effective beyond the directly simulated trajectory.

*Table 8.* Impact of simulation trajectory length ($M$).

| Trajectory ($M$) | 1 | 3 | 5 | 10 |
|---|---|---|---|---|
| ASR (%) | 65.2 | 82.4 | **86.5** | 86.5 |

**Proxy Dataset Size and Hyperparameter Sensitivity.** Since PBTO relies on a local proxy dataset rather than the victim's private data, the number of proxy images directly affects its practical applicability. We vary the proxy data size from 1,000 to 10,000 images per class on CIFAR-100 and report ASR after the victim completes ten CIL tasks.

Table 9 shows that PBTO reaches 76.8% ASR with 1,000 proxy images per class. Increasing the proxy set to 5,000 images improves ASR to 86.5%, while further increasing it to 10,000 images yields only a small gain to 87.1%. This saturation trend suggests that PBTO does not require access to the victim's full training set to identify useful structural patterns. Appendix C provides additional analyses of the replay buffer size $\mathcal{M}$, the iterative refinement budget $T_{\max}$, and the representation-alignment weight $\lambda$ for $\mathcal{L}_s$. Appendix H discusses the computational cost; on CIFAR-100, trigger optimization requires about 2.2 GPU hours on an RTX 4090.

### 5.7. Mechanism: Feature Space Visualization

To further diagnose why PBTO remains effective after continual updates, we visualize the penultimate-layer feature space after sequentially learning ten tasks. If the optimized trigger only acted as an isolated shortcut, subsequent task learning would tend to decouple poisoned samples from benign target-class representations; as this association is overwritten, poisoned samples would drift away from the target-class manifold and the attack would lose effectiveness. Therefore, a persistent attack should keep poisoned samples embedded near the benign target-class manifold even after the shared feature extractor is repeatedly updated.

Figure 4 shows that PBTO poisoned samples remain close to

*Table 9.* Impact of proxy data size per class on CIFAR-100.

| Size | 1,000 | 2,000 | **5,000** | 10,000 |
|---|---|---|---|---|
| ASR (%) | 76.8 | 82.4 | **86.5** | 87.1 |

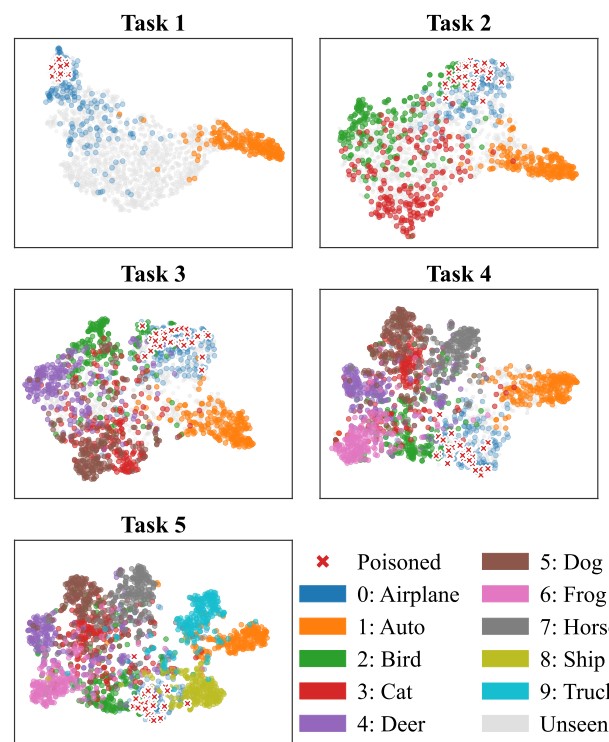

*Figure 4.* Feature space visualization (t-SNE) on CIFAR-10 after five CIL tasks. Red points denote poisoned samples, and blue points denote benign target-class samples.

the benign target-class cluster rather than separating into an isolated trigger-specific region. This feature-space structure provides supporting visual evidence for PBTO's structural anchoring strategy: the trigger-target association is embedded into target-class structural representations that remain useful during continual learning, which helps explain the sustained ASR observed in the main and task-wise results.

## 6. Conclusion

In this paper, we proposed PBTO, a persistent targeted backdoor attack for class-incremental learning. We first show that task-critical neurons are not stable under continuous CIL updates. We then examine how CIL models retain knowledge when learning semantically related tasks. Our analysis shows that CIL models mainly preserve such knowledge in shallow and structurally invariant subspaces, while deeper representations tend to drift across tasks. PBTO exploits this finding by simulating future parameter trajectories with proxy tasks and optimizing a universal trigger that maintains high ASR across benchmarks and architectures, clearly outperforming existing baselines.

## Impact Statement

This paper studies persistent backdoor attacks in class-incremental learning and therefore has dual-use implications. The proposed analysis and attack methodology could be misused to compromise continually updated models. Our goal is to expose an underexplored security risk, provide reproducible evidence for evaluating CIL robustness, and motivate defenses that account for structural trigger persistence. We encourage the use of these results for security assessment and mitigation rather than for deployment against real systems.

## Acknowledgements

Jianting Ning is supported by the Science Foundation of Zhejiang Sci-Tech University (ZSTU) under Grant No. 26222227-Y and the Program of Zhejiang Key Laboratory of Digital Fashion and Data Governance (No. 2024E10049).

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

## A. Detailed Experimental Setup

### A.1. Hybrid Proxy Dataset Construction

As mentioned in Section 5.1, PBTO constructs a local proxy dataset $\tilde{\mathcal{D}}$ to simulate the parameter changes induced by CIL. This process does not access the victim's private training data or the data from future tasks.

Specifically, we use two data sources:

1. **Public Repositories:** We collect public images using class names or semantically related queries. We then keep samples that match the target-class semantics after manual or classifier-assisted filtering.

2. **Synthetic Generation (DDPM):** When public images are insufficient or lack intra-class diversity, we use a pre-trained Denoising Diffusion Probabilistic Model (DDPM) (Ho et al., 2020) to generate additional samples.

For our standard experiments, each proxy class contains 5,000 images, and the proxy set is balanced across classes. The proxy set is used only for local surrogate trajectory simulation and trigger optimization. It does not contain the victim's private training samples or data from future CIL tasks.

## B. Additional Subspace Stability Metrics

Following the metrics introduced in Sec. 3.3, this appendix provides the full temporal and cross-architecture subspace stability measurements. The main text reports a compressed ResNet-18/CIFAR-100 summary, while this appendix expands the analysis across different task stages and architectures. These results verify that the shallow-to-deep stability pattern is consistent across CKA, Grassmann distance, and projection-based variance retention, rather than being an artifact of a single metric, time point, or model architecture.

*Table 10.* Grassmann distance of each ResNet-18 module on CIFAR-100. Smaller values indicate more stable subspaces.

| Module | Task 1→2 | Task 1→5 | Task 1→10 |
|---|---|---|---|
| Module 1 | 0.03 | 0.06 | 0.08 |
| Module 2 | 0.05 | 0.08 | 0.12 |
| Module 3 | 0.18 | 0.29 | 0.38 |
| Module 4 | 0.25 | 0.36 | 0.45 |

Tables 10 and 11 report the ResNet-18 results on CIFAR-100. After ten tasks, Module 1 has a Grassmann distance of only 0.08 and retains 0.92 of the initial-task variance. In contrast, Module 4 reaches a Grassmann distance of 0.45 and retains only 0.47 variance. Both metrics show the same trend: shallow modules are more stable, while deeper modules drift more clearly during continual learning.

*Table 11.* Projection-based variance retention of each ResNet-18 module on CIFAR-100. Larger values indicate higher preservation of the initial-task subspace.

| Module | Task 1→2 | Task 1→5 | Task 1→10 |
|---|---|---|---|
| Module 1 | 0.98 | 0.95 | 0.92 |
| Module 2 | 0.96 | 0.93 | 0.88 |
| Module 3 | 0.82 | 0.68 | 0.54 |
| Module 4 | 0.74 | 0.58 | 0.47 |

Table 12 further compares ResNet-18 and VGG-16. Both CNN architectures show higher CKA in shallow layers and lower CKA in deep layers. For example, from Task 1 to Task 10, ResNet-18 has a shallow CKA of 0.91 and a deep CKA of 0.49, while VGG-16 has a shallow CKA of 0.89 and a deep CKA of 0.45. This indicates that the shallow-stable pattern is not unique to ResNet-18.

Tables 13 and 14 report the results on ViT-Small/16. From shallow to deep layer groups, CKA decreases from 0.84 to 0.39, while Grassmann distance increases from 0.13 to 0.51. The shallow group of ViT is slightly less stable than the shallow CNN modules, which is consistent with the lower ASR observed in the ViT experiments. Nevertheless, the shallow group is still clearly more stable than the deeper transformer blocks and can serve as a usable anchor for PBTO.

*Table 12.* Cross-architecture CKA similarity on CIFAR-100. Similarity is measured between Task 1 representations and later task states on unpoisoned CIL models.

| Architecture | Layer Group | Task 1→2 | Task 1→5 | Task 1→10 |
|---|---|---|---|---|
| ResNet-18 | Shallow | 0.97 | 0.94 | 0.91 |
| ResNet-18 | Deep | 0.72 | 0.58 | 0.49 |
| VGG-16 | Shallow | 0.95 | 0.92 | 0.89 |
| VGG-16 | Deep | 0.68 | 0.54 | 0.45 |

*Table 13.* CKA similarity on ViT-Small/16. The model is divided into four layer groups from shallow to deep.

| Layer Group | Task 1→2 | Task 1→5 | Task 1→10 |
|---|---|---|---|
| Group 1 (Shallow) | 0.94 | 0.88 | 0.84 |
| Group 2 | 0.87 | 0.76 | 0.68 |
| Group 3 | 0.75 | 0.61 | 0.52 |
| Group 4 (Deep) | 0.63 | 0.48 | 0.39 |

*Table 14.* Grassmann distance on ViT-Small/16. Smaller values indicate more stable subspaces.

| Layer Group | Task 1→2 | Task 1→5 | Task 1→10 |
|---|---|---|---|
| Group 1 (Shallow) | 0.05 | 0.09 | 0.13 |
| Group 2 | 0.11 | 0.18 | 0.25 |
| Group 3 | 0.19 | 0.30 | 0.39 |
| Group 4 (Deep) | 0.27 | 0.40 | 0.51 |

## C. Extended Ablation Studies

This section reports the detailed numerical results for the sensitivity analyses discussed in Section 5.6.

### C.1. Data Efficiency and Buffer Size Tables

Table 9 in the main paper reports the results for proxy data size. Table 15 further shows the effect of memory buffer size $\mathcal{M}$.

*Table 15.* Effect of memory buffer size ($\mathcal{M}$) on the final ASR on CIFAR-100. Results are reported after Task 10.

| Method | Buffer Size ($\mathcal{M}$) | | | |
|---|---|---|---|---|
| | 500 | 1,000 | 2,000 | 5,000 |
| BadNets | 1.2% | 2.5% | 4.5% | 12.8% |
| WaNet | 3.8% | 4.4% | 3.2% | 10.5% |
| LTB | 3.6% | 3.8% | 4.7% | 6.2% |
| **Ours** | **10.4%** | **69.2%** | **86.5%** | **91.3%** |

As shown in Table 15, a smaller replay buffer makes persistence more difficult for all attacks. When $\mathcal{M} = 500$, PBTO obtains 10.4% final ASR, which is still higher than the baselines but is clearly limited by the small buffer. As the buffer size increases, PBTO gradually improves and reaches 86.5% and 91.3% ASR when $\mathcal{M} = 2,000$ and $5,000$, respectively. In contrast, BadNets, WaNet, and LTB remain at much lower final ASR under the same settings.

This indicates that the memory buffer size affects how well the backdoor association can be preserved in CIL. A larger buffer mitigates forgetting and makes it easier for PBTO to maintain the trigger-target association. Under strict buffer constraints, PBTO still behaves more stably than static or single-stage baselines.

### C.2. Impact of Iterative Refinement and $\lambda$

Figure 5 shows the effects of iterative refinement and the representation-alignment weight $\lambda$. In the left plot, ASR increases quickly in the first several refinement rounds and becomes stable after about 50 rounds. This suggests that re-simulating the poisoned trajectory helps reduce the mismatch between the initial clean trajectory and the actual attack process. Once the

trigger has adapted to the main trajectory changes, additional rounds provide limited gains.

The right plot reports the sensitivity to $\lambda$. The best result is obtained around $\lambda = 1.0$. When $\lambda$ is too small, the representation-alignment constraint is weak and the trigger is more likely to exploit short-term features of the current model state. When $\lambda$ is too large, the optimization overemphasizes target-class representation alignment and weakens the trigger's discriminative effect. We therefore set $\lambda = 1.0$ in the standard setting.

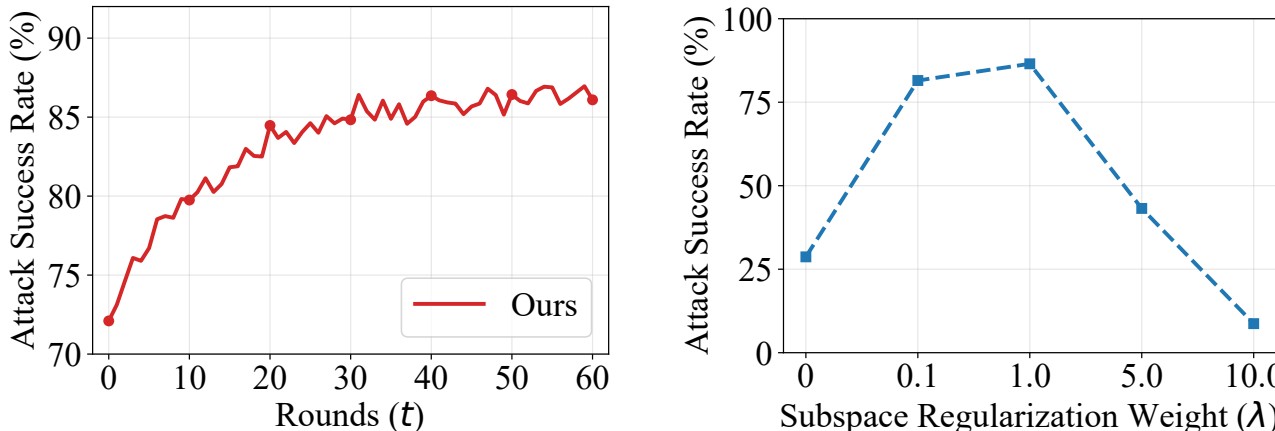

*Figure 5.* Left: impact of refinement rounds on ASR. Right: impact of the representation-alignment weight $\lambda$ on ASR.

## D. Additional Baseline Comparisons

In addition to the main experiments, we include ISSBA (Li et al., 2021a), LIRA (Doan et al., 2021), and WaveAttack (Xia et al., 2024) as additional baselines on CIFAR-100. These methods represent stealthy attacks that can achieve high ASR in static or single-stage learning settings. However, they do not explicitly account for the representation drift caused by continuous parameter updates in class-incremental learning. This distinction matters for persistent attacks: a trigger may be effective immediately after poisoning, but later classes can still weaken it as they repurpose the shared feature extractor.

*Table 16.* Additional baseline comparison on CIFAR-100. Results are reported on the final model after ten CIL tasks.

| Attack | BA (%) | ASR (%) |
|---|---|---|
| BadNets | 68.5±0.7 | 4.5±0.5 |
| WaNet | 68.1±0.6 | 3.2±1.1 |
| DRUPE | 67.9±0.8 | 14.2±1.9 |
| LTB | 67.5±1.2 | 4.7±3.2 |
| ISSBA | 68.3±0.6 | 4.1±1.2 |
| LIRA | 68.0±0.7 | 2.8±1.3 |
| WaveAttack | 68.2±0.7 | 7.6±1.5 |
| **PBTO (Ours)** | 67.2±0.6 | **86.5**±0.9 |

As shown in Table 16, after all ten CIL tasks are completed, ISSBA, LIRA, and WaveAttack obtain final ASRs of 4.1%, 2.8%, and 7.6%, respectively. This is consistent with the degradation observed for BadNets, WaNet, DRUPE, and LTB. In contrast, PBTO keeps 86.5% ASR on the final model while maintaining a similar BA level.

The result also shows that the failure of the baselines is not specific to visible patch triggers. Even stealthier triggers such as those used by ISSBA, LIRA, and WaveAttack are difficult to preserve after continual CIL updates. Thus, the main limitation of these methods in CIL is not a clear drop in clean accuracy, but the lack of a mechanism for maintaining the trigger-target association across later task updates.

# E. Scalability and Architecture Generalization

This section examines whether PBTO depends on the specific ResNet-18/CIFAR setting. We consider three additional cases: a larger and higher-resolution dataset, VGGFace2 (Cao et al., 2018); a same-architecture ViT setting, where the surrogate and victim both use ViT-Small/16; and cross-architecture transfer, where the trigger is optimized on a ResNet-18 surrogate and evaluated on different victim architectures.

*Table 17.* Scalability experiment on VGGFace2. Results are reported after ten CIL tasks. VGGFace2 contains 3.31M images and 9,131 identities at 224×224 resolution.

| Attack | BA (%) | ASR (%) |
|---|---|---|
| BadNets | 74.8±0.5 | 5.3±1.1 |
| WaNet | 74.5±0.6 | 3.7±0.9 |
| LTB | 74.3±0.7 | 6.1±1.3 |
| **PBTO (Ours)** | 74.1±0.5 | **85.9**±1.0 |

*Table 18.* Same-architecture ViT evaluation on CIFAR-100. The PBTO trigger is optimized on a ViT-Small/16 surrogate and evaluated on a ViT-Small/16 victim after ten CIL tasks. All attacks poison only Task 1, and results are averaged over three runs.

| Attack | BA (%) | ASR (%) |
|---|---|---|
| BadNets | 64.2±0.7 | 3.8±0.8 |
| WaNet | 63.9±0.8 | 2.5±0.7 |
| LTB | 63.6±0.9 | 3.1±0.8 |
| **PBTO (Ours)** | 63.5±0.7 | **81.4**±1.3 |

*Table 19.* Cross-architecture transfer experiment. Triggers are optimized on a ResNet-18 surrogate and evaluated after the victim completes CIL.

| Target Model | BA (%) | LTB ASR (%) | PBTO ASR (%) |
|---|---|---|---|
| VGG-16 | 82.0±0.5 | 5.5±1.8 | **78.1**±1.2 |
| DenseNet-121 | 82.2±0.6 | 6.3±2.1 | **82.2**±0.9 |
| MobileNetV2 | 81.8±0.5 | 5.8±1.9 | **81.5**±1.0 |
| ViT-Small/16 | 76.3±0.8 | 4.1±1.5 | **68.7**±1.8 |

Table 17 reports the results on VGGFace2. This dataset contains 3.31M images from 9,131 identities at $224 \times 224$ resolution. After ten CIL tasks, BadNets, WaNet, and LTB obtain final ASRs of 5.3%, 3.7%, and 6.1%, respectively, while PBTO reaches 85.9%. This indicates that the persistence effect of PBTO is not limited to smaller benchmarks such as CIFAR-100 or Tiny-ImageNet.

Table 18 isolates the same-architecture ViT setting from cross-architecture transfer. Here, the surrogate trajectory and victim backbone both use ViT-Small/16 (Dosovitskiy et al., 2021) on CIFAR-100, while the poison-only protocol remains unchanged: the attacker poisons only Task 1, and ASR and BA are measured after all ten CIL tasks. PBTO reaches 81.4% ASR with a BA similar to the baselines, whereas BadNets, WaNet, and LTB remain below 4% ASR. This confirms that PBTO can exploit stable structural representations in a transformer victim, not only in CNN backbones.

Table 19 further reports the cross-architecture transfer results. When the trigger is optimized on ResNet-18 and transferred to VGG-16, DenseNet-121, and MobileNetV2, PBTO obtains final ASRs of 78.1%, 82.2%, and 81.5%, respectively. In the CNN-to-ViT-Small/16 setting, ASR drops from 81.4% in the same-architecture ViT setting to 68.7%. This drop is expected because the surrogate and victim have more divergent feature operators. Under the same setting, LTB obtains only 4.1% ASR.

Together, these results indicate that PBTO does not simply memorize the parameters or structure of ResNet-18. The same-architecture ViT result shows that PBTO remains effective on a transformer backbone, while the cross-architecture results show that architectural mismatch decreases ASR but does not eliminate persistence.

# F. Task-wise Persistence Analysis

Table 20 provides the per-task ASR analysis referenced in Sec. 5.2. This table checks whether PBTO's persistence is only an endpoint effect. PBTO maintains high ASR throughout the ten-task CIL sequence, decreasing gradually from 98.0% after Task 1 to 86.5% after Task 10. In contrast, the baselines lose the trigger-target association quickly, and the strongest baseline reaches only 14.2% ASR by Task 10.

*Table 20.* Task-wise ASR on CIFAR-100 across ten CIL tasks. The trigger is injected only in Task 1 and evaluated after each subsequent task.

| Attack | T1 | T2 | T3 | T4 | T5 | T6 | T7 | T8 | T9 | T10 |
|---|---|---|---|---|---|---|---|---|---|---|
| BadNets | 95.2 | 48.3 | 27.5 | 17.8 | 12.6 | 9.4 | 7.5 | 6.2 | 5.3 | 4.5 |
| WaNet | 93.4 | 39.6 | 22.1 | 14.5 | 10.2 | 7.3 | 5.6 | 4.5 | 3.8 | 3.2 |
| DRUPE | 94.8 | 58.6 | 40.3 | 31.5 | 25.7 | 21.8 | 19.1 | 17.0 | 15.4 | 14.2 |
| LTB | 88.6 | 45.2 | 28.7 | 19.3 | 13.8 | 10.1 | 7.8 | 6.4 | 5.5 | 4.7 |
| **PBTO** | **98.0** | **95.8** | **94.2** | **93.1** | **91.9** | **90.7** | **89.6** | **88.5** | **87.5** | **86.5** |

# G. Robustness to Surrogate Mismatch

In practice, the attacker usually does not know the future task order, the victim data distribution, or the exact training hyperparameters. We therefore test the sensitivity of PBTO to surrogate-victim mismatch. Tables 21, 22, and 23 report the detailed results under task-ordering, cross-dataset, and training-hyperparameter mismatches.

*Table 21.* Final ASR and BA under different task orderings on CIFAR-100.

| Task Ordering | Final ASR (%) | BA (%) |
|---|---|---|
| Ordering #1 | 86.5 | 67.2 |
| Ordering #2 | 85.3 | 67.8 |
| Ordering #3 | 87.1 | 66.5 |
| Ordering #4 | 84.6 | 67.5 |
| Ordering #5 | 86.8 | 66.9 |
| **Mean $\pm$ Std.** | **86.1$\pm$1.0** | **67.2$\pm$0.5** |

*Table 22.* Cross-dataset surrogate evaluation. The trigger is optimized with a surrogate trained on the source dataset and evaluated after the victim completes ten CIL tasks.

| Surrogate Data | Victim Data | BA (%) | ASR (%) |
|---|---|---|---|
| CIFAR-100 | CIFAR-100 | 67.2$\pm$0.6 | 86.5$\pm$0.9 |
| Tiny-ImageNet | CIFAR-100 | 66.8$\pm$0.7 | 72.4$\pm$1.6 |
| CIFAR-100 | Tiny-ImageNet | 57.1$\pm$1.8 | 69.3$\pm$1.9 |

*Table 23.* Surrogate-victim training-hyperparameter mismatch on CIFAR-100. The surrogate setting is fixed while the victim training setting varies.

| Surrogate LR | Victim LR | Surrogate BS | Victim BS | ASR (%) |
|---|---|---|---|---|
| 0.01 | 0.01 | 128 | 128 | 86.5$\pm$0.9 |
| 0.01 | 0.005 | 128 | 128 | 87.2$\pm$0.8 |
| 0.01 | 0.05 | 128 | 128 | 83.1$\pm$1.2 |
| 0.01 | 0.01 | 128 | 64 | 85.8$\pm$1.0 |
| 0.01 | 0.01 | 128 | 256 | 86.1$\pm$0.9 |

As shown in Table 21, across five random task orders, PBTO obtains final ASRs from 84.6% to 87.1%, with an average of 86.1 $\pm$ 1.0%. This indicates that the trigger is not tied to a specific class ordering.

The cross-dataset surrogate setting is more difficult. As shown in Table 22, when both the surrogate and victim use CIFAR-100, the ASR is 86.5%. When the surrogate uses Tiny-ImageNet and the victim uses CIFAR-100, the ASR drops to

72.4%. When the surrogate uses CIFAR-100 and the victim uses Tiny-ImageNet, the ASR is 69.3%. This drop indicates that distribution mismatch weakens the quality of the surrogate trajectory approximation, but PBTO still retains a non-trivial ASR under this mismatch.

Table 23 shows that learning-rate and batch-size mismatches have a smaller effect. When the victim learning rate is changed from 0.01 to 0.005 or 0.05, or when the victim batch size is changed from 128 to 64 or 256, the final ASR remains between 83.1% and 87.2%. Overall, PBTO does not require the surrogate trajectory to exactly match the victim trajectory, but data distribution mismatch causes a more visible performance drop.

## H. Computational Cost and Potential Countermeasures

PBTO optimizes the trigger once during the offline attack preparation stage. Table 24 reports the optimization time on CIFAR-100. The extra cost mainly comes from evaluating and updating the trigger across multiple surrogate trajectory checkpoints. Compared with 1.8 GPU hours for LIRA and 1.9 GPU hours for ISSBA, PBTO takes 2.2 GPU hours. It is slightly more expensive, but remains in the same order of magnitude.

*Table 24.* Trigger optimization time on CIFAR-100. Runtime is measured in GPU hours.

| Attack | Time (GPU hours) |
|---|---|
| LIRA | 1.8 |
| ISSBA | 1.9 |
| **PBTO (Ours)** | **2.2** |

This result shows that PBTO introduces additional offline computation, but the cost is not incurred during victim inference. Since the trigger only needs to be optimized once and can then be reused, this cost mainly affects the attack preparation stage. Checkpoint subsampling, gradient accumulation, or early stopping may further reduce the cost of trigger refinement. We leave a more systematic efficiency study to future work.

From the defense perspective, PBTO is difficult to remove directly because it uses shallow structural features that are also useful for benign recognition. Directly disrupting these features may reduce clean accuracy or worsen catastrophic forgetting. Based on this mechanism, there are several possible defense directions. First, shallow-feature randomization may weaken the structural anchor, but its effect on normal representations needs to be controlled. Second, cross-layer consistency analysis may help detect abnormal relations between stable shallow features and drifting deep semantics. Third, training-time data sanitization may filter poisoned samples before they enter the CIL stream. These directions require more systematic evaluation, and we do not treat them as solved by this work.

