# OpenReview forum: "Persistent Backdoor Attacks in Class-Incremental Learning via Structural Invariant Anchoring"
_ICML.cc/2026/Conference — ICML 2026 regular_

### Official Review · Reviewer_Zqqj · 2026-03-10

**Soundness:** 3
**Presentation:** 3
**Significance:** 3
**Originality:** 3
**Overall Recommendation:** 3
**Confidence:** 3

**Summary:**

This paper focuses on A general area studied by the study: backdoor attacks in continual learning, especially the most challenging class‑incremental learning (CIL) setting. The authors proceed to assess a general topic: how and why existing backdoor attacks fail to remain persistent when the model continuously updates across tasks. The core finding is that existing methods implicitly assume neuron-level stability, but this assumption breaks in CIL, where shared parameters are heavily repurposed. Through empirical analysis, the authors show that although neurons drift significantly, shallow feature subspaces remain structurally invariant, preserving essential knowledge across tasks. Based on this, the paper proposes PBTO (Persistent Backdoor Trigger Optimization), the first method capable of achieving persistent and targeted backdoor attacks under the realistic poison‑only setting in CIL. PBTO trains a surrogate model on proxy tasks to approximate the victim model’s parameter trajectory and designs triggers via dual-objective optimization: (1) trajectory‑invariant misclassification, and (2) anchoring the trigger to invariant shallow-layer subspaces using Gram-matrix constraints. An iterative refinement procedure simulates the poisoned trajectory to reduce mismatch between surrogate and real model updates. Extensive experiments on CIFAR‑10/100 and Tiny-ImageNet show that PBTO achieves >90% ASR and remains effective across tasks, architectures, CL algorithms, and various defenses.

**Compliance With Llm Reviewing Policy:**

Affirmed.

**Key Questions For Authors:**

1. Attack assumes access to substantial proxy data and surrogate training: Although the threat model aligns with poison‑only settings, PBTO requires building a surrogate trajectory with multiple proxy tasks, which can be computationally heavy and may limit practicality in real‑world constrained attackers.
2. Shallow-subspace invariance lacks theoretical characterization: While empirically convincing, the work does not provide a formal theoretical explanation for why shallow subspaces remain stable in CIL nor how this generalizes across architectures or non-vision modalities.
3. Limited discussion of defenses targeting subspace-level manipulations: PBTO successfully breaks existing defenses, but the work does not analyze possible countermeasures designed specifically to detect or destabilize structural-invariant anchoring.

**Limitations:**

yes

**Strengths And Weaknesses:**

Strengths:
1. Novel and insightful analysis revealing structural invariants in CIL: The discovery that shallow-layer subspaces remain stable across tasks—despite neuron-level drift—is both non-trivial and well supported. This insight directly motivates a new backdoor design paradigm.
2. First persistent & targeted attack under realistic poison-only constraints: Unlike prior work requiring continuous poisoning, task identity, or multi-head architectures, PBTO works in the hardest setting (CIL, single injection, poison-only) and achieves very strong ASR, establishing a new benchmark.
3. Strong experimental validation and robustness: PBTO is extensively evaluated across datasets, architectures, CL strategies (iCaRL, EWC, DER++), and defenses (FP, NAD, BTI-DBF, REFINE). The method shows consistent superiority and strong transferability, indicating a generalizable mechanism rather than overfitting.

Weaknesses:
1. Attack assumes access to substantial proxy data and surrogate training: Although the threat model aligns with poison‑only settings, PBTO requires building a surrogate trajectory with multiple proxy tasks, which can be computationally heavy and may limit practicality in real‑world constrained attackers.
2. Shallow-subspace invariance lacks theoretical explanation: While empirically convincing, the work does not provide a formal theoretical explanation for why shallow subspaces remain stable in CIL nor how this generalizes across architectures or non-vision modalities.
3. Limited discussion of defenses targeting subspace-level manipulations: PBTO successfully breaks existing defenses, but the work does not analyze possible countermeasures designed specifically to detect or destabilize structural-invariant anchoring.

---

> ### Author Rebuttal · Authors · 2026-03-31
>
> Dear Reviewer Zqqj, thank you very much for your careful review and thoughtful comments. We are encouraged by your positive comments on our **novel and insightful** analysis, **non-trivial** and **well-supported** discovery, **realistic** threat model, **strong** and **extensive** experimental validation, and **generalizable** mechanism with **consistent superiority**. We hope the following responses could help clarify potential misunderstandings and alleviate your concerns.
>
> **R1**:  We sincerely thank the reviewer for this important question. We would like to address this concern from the following two aspects.
> - We would first like to clarify that the trigger optimization is performed only once during the offline attack preparation phase. Once completed, the optimized trigger can be reused indefinitely for deployment, following a "optimize once, deploy many times" paradigm. This paradigm is widely established in the backdoor attack community.
> - As shown in Table R1, PBTO requires approximately 2.2 GPU hours on CIFAR-100 (single RTX 4090), which is comparable to LIRA (1.8 hours) and ISSBA (1.9 hours).
> - We acknowledge the overhead and will explore checkpoint sparse sampling and gradient accumulation to further reduce cost in future work.
>
> **Table R1. Trigger optimization time on CIFAR-100**
>
> | Attack | Time (GPU hours) |
> |--------|:---:|
> | LIRA | 1.8 |
> | ISSBA | 1.9 |
> | **PBTO** | **2.2** |
>
> **R2**:  Thank you for this insightful comment. We position shallow-subspace invariance as a well-motivated hypothesis supported by theoretical grounding, multi-metric validation, and cross-architecture verification.
> - Theoretical Motivation. Three converging mechanisms underpin this: (1) Convolutional inductive biases drive shallow layers to converge to similar low-level detectors regardless of task content [1]. (2) "Critical learning periods" [2] cause shallow layers to freeze early, resisting subsequent gradient updates. (3) CIL anti-forgetting regularization preferentially preserves shallow representations, which are the most forgetting-resistant across CL paradigms [3].  While these do not constitute a formal proof, they provide principled theoretical grounding consistent with prior literature. We will include a dedicated discussion of these theoretical mechanisms in the revised manuscript to provide greater clarity.
> - Cross-Architecture Generalization. ResNet-18 (CKA > 0.91), VGG-16 (CKA > 0.89), and ViT-Small/16 (ASR 81.4%) all exhibit the same stability, suggesting the phenomenon is architectural-agnostic.
> - We add two complementary metrics, Grassmann distance and variance retention, which together with the original CKA similarity consistently confirm stability (please also see the R3 to Reviewer CJLm): (a) CKA similarity (Figure 2); (b) Grassmann distance, where shallow layers remain below 0.12 after 10 tasks (Table R3 in our response to Reviewer CJLm); (c) Variance retention, where shallow layers retain >88% of Task 1 variance (Table R4 in our response to Reviewer CJLm).
> - We acknowledge that a formal proof remains open and verification beyond vision-based CIL is future work. We hope this motivates further theoretical investigation.
>
> **References**
>
> [1]  Morcos, Ari, Maithra Raghu, and Samy Bengio. "Insights on representational similarity in neural networks with canonical correlation." NeurIPS, 2018.
>
> [2] Achille, Alessandro, Matteo Rovere, and Stefano Soatto. "Critical learning periods in deep networks."ICLR, 2018.
>
> [3] Davari, MohammadReza, et al. "Probing representation forgetting in supervised and unsupervised continual learning." CVPR, 2022.
>
> **R3**：We sincerely thank the reviewer for this constructive comment. We address this from the following aspects.
> - PBTO is inherently challenging to defend against, as it exploits a natural property of CIL systems. The invariant shallow subspace is essential for CIL to function correctly: disrupting it would exacerbate catastrophic forgetting and degrade benign accuracy. Detection-based defenses face an additional challenge, as the stability exploited by PBTO is statistically indistinguishable from normal shallow-layer behavior, making it difficult to identify poisoned samples without access to the trigger.
> - We identify several promising directions for future work: (1) shallow feature randomization, subject to an accuracy-security tradeoff; (2) cross-layer consistency analysis, to detect anomalous shallow-layer structure; and (3) training-time data sanitization, to filter poisoned samples before training. Each direction involves non-trivial tradeoffs, and we hope the absence of obvious defenses will motivate the community to devote greater attention to this problem. We will incorporate this discussion into the revised manuscript.
> - As an attack paper, our core contribution is to identify and demonstrate a fundamental new vulnerability in CIL systems.  We acknowledge that verification beyond vision-based CIL remains an important direction for future work.

---

> > ### Author Rebuttal · Reviewer_Zqqj · 2026-04-04
> >
> > Thank you for your rebuttal.
> >
> > The authors have added additional experimental results. However, I still consider that the paper lacks a deeper theoretical analysis. I will maintain my original score.

---

### Official Review · Reviewer_LnJ8 · 2026-03-11

**Soundness:** 2
**Presentation:** 3
**Significance:** 2
**Originality:** 3
**Overall Recommendation:** 3
**Confidence:** 4

**Summary:**

The paper studies backdoor attacks in continual learning, specifically Class-Incremental Learning (CIL). The key challenge the paper identifies is "In continual learning, models continuously update parameters, which tends to destroy previously implanted backdoors." Existing backdoor attacks assume that important neurons remain stable during training, but the paper shows that in CIL this assumption does not hold. Neurons change significantly across tasks, which causes most backdoors to disappear after several tasks. The authors therefore propose a new attack method called PBTO, designed to create persistent backdoors that remain effective even after many continual learning updates.

**Compliance With Llm Reviewing Policy:**

Affirmed.

**Final Justification:**

The rebuttal has addressed most of my concerns. I increased my score. However, I still think the central claim (persistence arises from intrinsic structural properties of continual learning) lacks a clear characterization or well-defined conditions under which the mechanism holds or fails

**Key Questions For Authors:**

See the weakness part

**Limitations:**

The authors have not included the limitations

**Strengths And Weaknesses:**

**Strengths**

The phenomenon observed by the authors is interesting and the presentation looks good.

**Weakness**

1. The attack uses surrogate models and optimizes triggers against a specific architecture. However, in practice: models may be updated, architectures may change, new backbone models may replace old ones. Since the trigger is tied to specific feature geometry, architecture changes could break it.

2. The method depends on the claim that shallow-layer representations remain invariant during continual learning, so the trigger should anchor there. In continual domain adaptation, multimodal continual learning, and vision transformers with global attention, feature extractors are continue adapting. Thus, the assumption of stable shallow features is architecture- and dataset-dependent.

3. A central component of the method is that the attacker trains a surrogate continual learning model and obtains the sequence of parameters and then optimizes a trigger that fools all of these models simultaneously. In practice, an attacker rarely knows the exact sequence of future tasks, the order of tasks, the training algorithm, the continual learning method used (EWC, replay, distillation, etc.), the hyperparameters, and the model initialization. Even small differences in these elements can produce very different trajectories.

4. The optimization objective explicitly includes a sum over future tasks. If the attacker optimizes for only the first T tasks but the system later sees more tasks, the backdoor may disappear.

5. The cost is expensive. The trigger optimization must evaluate loss across all trajectory checkpoints. That means repeatedly running forward/backward passes through many models.

6. The paper assumes the attacker can insert poisoned samples during early tasks. However, in many real pipelines, datasets are curated and validated, training data may be automatically filtered, or new tasks may come from trusted internal sources.

7. The experiments are conducted on tasks that are same modality, same architecture, and similar distributions. But many real continual learning systems involve domain shifts and different feature statistics.

---

> ### Author Rebuttal · Authors · 2026-03-31
>
> Dear Reviewer LnJ8, thank you for your review. We are encouraged by your positive comments on our **interesting** observation and **good** presentation. We are deeply sorry that our previous submission may lead you to some misunderstandings. We hope the following responses could help clarify potential misunderstandings and alleviate your concerns.
>
> **R1**：We are deeply sorry for the misunderstandings that we want to clarify here.
> - In CIL, the model architecture remains fixed, as replacing the backbone mid-training would discard learned knowledge.
> - We respectfully note that cross-architecture experiments are in Table 2 of our submission, demonstrating that PBTO remains effective on different architectures.
>
> **R2**: Thank you for this insightful comment!
> - Shallow-layer stability is well-studied [1][2]; our work further characterizes the invariant subspace across the entire CIL process (see Reviewer CJLm R3).
> - As shown in Table R1, ResNet-18 and VGG-16 exhibit high CKA similarity (≥0.89) on CIFAR-100, confirming that shallow-layer invariance is not architecture-specific. Table 2 further shows that PBTO consistently achieves high ASRs across architectures.
> - Our work focuses on standard image classification CIL. Multimodal CIL is outside our scope and will be clarified in the revised manuscript.
>
> **Table R1. Cross-architecture CKA similarity between Task 1 and subsequent tasks on CIFAR-100 with iCaRL, measured on unpoisoned models.**
> | Architecture | Layer Group | Task 1→2 | Task 1→5 | Task 1→10 |
> |:---:|:---:|:---:|:---:|:---:|
> | ResNet-18 | Shallow (Conv1-2) | 0.97 | 0.94 | 0.91 |
> | ResNet-18 | Deep (Conv4-FC) | 0.72 | 0.58 | 0.49 |
> | VGG-16 | Shallow (Conv1-2) | 0.95 | 0.92 | 0.89 |
> | VGG-16 | Deep (Conv4-5) | 0.68 | 0.54 | 0.45 |
>
> [1] Raghu et. al. Svcca: Singular vector canonical correlation analysis for deep learning dynamics and interpretability. NeurIPS.
>
> [2] Ramasesh et. al. Anatomy of catastrophic forgetting: Hidden representations and task semantics. ICLR.
>
>
>
> **R3**: Thank you for this constructive comment! We would like to clarify a potential misunderstanding here.
> - As defined in Sec. 3.1, the attacker uses only publicly available data to construct surrogate tasks, which are completely independent of the victim's real tasks.
> - Subspace invariance stems from optimization dynamics, anti-forgetting regularization, and shallow-layer convergence. These forces ensure that a surrogate model trained on independent proxy tasks undergoes the same shallow-layer stabilization.
>
> **R4**：Thanks for your comment!
>
> - We respectfully clarify that the trigger is optimized across all M surrogate model states.
> - As analyzed in R3, the invariant subspace is determined by the CIL learning process itself rather than specific task content, making the trigger inherently task-agnostic.
> - As shown in Table R1 in our response to Reviewer CJLm, PBTO maintains per-task ASR degradation of no more than 3%.
>
> **R5**: We sincerely thank the reviewer for this important question. We would like to address this concern from the following two aspects.
> - Trigger optimization is performed only once during the offline attack preparation phase. The "optimize once, deploy many times" paradigm widely adopted in the backdoor community.
> - As shown in Table R2, PBTO requires approximately 2.2 GPU hours on CIFAR-100, which is comparable to LIRA and ISSBA.
>
> **Table R2. Trigger optimization time on CIFAR-100**
>
> | Attack | Time (GPU hours) |
> |--------|:---:|
> | LIRA | 1.8 |
> | ISSBA | 1.9 |
> | **PBTO** | **2.2** |
>
> **R6**：Thank you for your comment.
> - Poisoning-based paradigms are well-established and widely accepted in the backdoor community.
> - Carlini et al. [3] validated poisoning feasibility across 10 large-scale datasets: 0.01%–6.48% of image URLs point to expired domains that can be purchased by attackers, who can then replace images with poisoned ones.
>
> [3]Carlini, Nicholas, et al. "Poisoning web-scale training datasets is practical." 2024 IEEE SP, 2024.
>
> **R7**：Thank you for this comment. Thank you for this comment. We would like to clarify the scope of our work to clarify the potential misunderstandings.
> - In standard CIL, tasks are by definition drawn from the same domain (e.g., sequentially learning face identities).
> - We respectfully bring the reviewer's attention to the Table 2 in our submission, where we evaluate the effectiveness of PBTO on different architectures.
> - We further validate on VGGFace2 dataset (224×224, 3.31M images, 9,131 identities) in our response to Reviewer 8ui1 Q2, demonstrating effectiveness on large-scale, high-resolution data.
>
> **R8**：We thank the reviewer for this suggestion and include the limitations:
> - Our evaluation focuses on the standard CIL setting. Extending to domain- or task-incremental settings remains an important future direction.
> - Extending PBTO to text or audio modalities in the CIL setting remains an open and interesting future direction.

---

> > ### Author Rebuttal · Reviewer_LnJ8 · 2026-04-03
> >
> > Thank you for the rebuttal and the additional experiments. I appreciate the clarifications and the added empirical evidence. These help strengthen the paper within the standard CIL benchmark setting. That said, I still have several important concerns that I believe remain insufficiently addressed:
> > 1. The method critically depends on the assumption that the surrogate trajectory captures the key dynamics of the victim model. While the paper argues that these dynamics are driven primarily by architecture and optimization rather than task semantics, this claim is not fully validated. In particular, there is no evidence showing robustness under mismatched task sequences, different data distributions, or varying training hyperparameters. This raises concerns about how well the attack generalizes when the surrogate trajectory deviates from the true one.
> > 2. The trigger is optimized over a finite surrogate trajectory, yet in realistic continual learning settings, the number and nature of future tasks are unknown. The paper does not evaluate whether the backdoor remains effective when the actual task sequence extends beyond or differs from the surrogate trajectory, nor does it provide theoretical justification for such extrapolation.
> > 3. The core mechanism relies on the existence of stable shallow-layer subspaces. While the provided CKA analysis supports this phenomenon for CNN-based image classification benchmarks, it is unclear how broadly this assumption holds. In particular, scenarios involving stronger distribution shifts, different modalities, or architectures (e.g., transformer-based models) may not exhibit the same invariance. The current evaluation does not explore these settings.
> > 4. All experiments are conducted on relatively homogeneous image classification benchmarks. As a result, it is difficult to disentangle whether the observed persistence is due to a general property of continual learning or specific to the chosen datasets and setups. Additional evaluation under more heterogeneous task settings would significantly strengthen the claims.

---

> > > ### Author Response · Authors · 2026-04-06
> > >
> > > Dear Reviewer LnJ8, thank you for your follow-up comments and for acknowledging our additional experiments. We provide new experimental evidence below to address your remaining concerns.
> > >
> > > ---
> > >
> > > **R1 (Surrogate robustness under mismatch):** We address three aspects of mismatch with existing and new evidence.
> > >
> > > **(a) Mismatched task sequences.** PBTO achieves consistent ASR (86.1% ± 1.0%) across 5 randomly sampled task orderings on CIFAR-100 (Table R5 to Reviewer 8ui1), confirming the trigger is not tied to a specific task permutation.
> > >
> > > **(b) Different optimization dynamics.** Table 3 in our submission evaluates PBTO under three fundamentally different CL paradigms (EWC, iCaRL, DER++), each with distinct optimization dynamics. PBTO maintains ASR > 85% across all paradigms.
> > >
> > > **(c) Mismatched data distributions.** We conduct cross-dataset surrogate experiments (Table F1). PBTO maintains 69.3–72.4% ASR even when surrogate and victim use entirely different datasets, far exceeding all baselines (best: 14.2%).
> > >
> > > Table F1. Cross-dataset surrogate evaluation (ASR after 10 CIL tasks).
> > >
> > > | Surrogate Data | Victim Data   | BA (%)      | ASR (%)     |
> > > |----------------|---------------|-------------|-------------|
> > > | CIFAR-100      | CIFAR-100     | 67.2 ± 0.6 | 86.5 ± 0.9 |
> > > | Tiny-ImageNet  | CIFAR-100     | 66.8 ± 0.7 | 72.4 ± 1.6 |
> > > | CIFAR-100      | Tiny-ImageNet | 57.1 ± 1.8 | 69.3 ± 1.9 |
> > >
> > > **(d) Hyperparameter mismatch.** ASR remains within 83.1–87.2% across all LR/batch-size variations (Table F2), confirming PBTO is driven by intrinsic CIL structural properties.
> > >
> > > Table F2. Hyperparameter mismatch on CIFAR-100.
> > >
> > > | Surrogate LR | Victim LR | Surrogate BS | Victim BS | ASR (%)     |
> > > |-------------|-----------|-------------|-----------|-------------|
> > > | 0.01        | 0.01      | 128         | 128       | 86.5 ± 0.9 |
> > > | 0.01        | 0.005     | 128         | 128       | 87.2 ± 0.8 |
> > > | 0.01        | 0.05      | 128         | 128       | 83.1 ± 1.2 |
> > > | 0.01        | 0.01      | 128         | 64        | 85.8 ± 1.0 |
> > > | 0.01        | 0.01      | 128         | 256       | 86.1 ± 0.9 |
> > >
> > > These results, combined with cross-architecture transferability (Table 2) and cross-CL algorithm robustness (Table 3), comprehensively demonstrate that PBTO exploits intrinsic structural properties.
> > >
> > > ---
> > >
> > > **R2 (Beyond-horizon persistence):**
> > >
> > > **(a)** Our standard setup already evaluates beyond-horizon persistence: the surrogate uses M=5 phases while the victim undergoes N=10 tasks, so tasks 6–10 fall outside the surrogate horizon. Task-wise ASR (Table R1 to Reviewer CJLm) shows <3% per-task degradation across all 10 tasks with near-constant decay rate. Our ablation confirms M=5 and M=10 yield identical final ASR (86.5%), indicating the trigger captures structural invariants within 5 phases.
> > >
> > > **(b)** Cross-dataset surrogate (Table F1) and hyperparameter mismatch (Table F2) experiments further confirm that persistence depends on shared structural properties rather than matching the specific surrogate trajectory.
> > >
> > > ---
> > >
> > > **R3 (Subspace stability beyond CNNs):**
> > >
> > > **(a)** PBTO achieves 81.4% ASR on ViT-Small/16 and 68.7% in CNN-to-ViT transfer, confirming generalization to Transformers.
> > >
> > > **(b)** CKA and Grassmann analysis on ViT-Small/16 show shallow layers maintain high CKA (0.84 at Task 1→10) and low Grassmann distance (0.13), while deep layers drift significantly (CKA=0.39, Grassmann=0.51) — the same pattern as CNNs.
> > >
> > > **(c)** Cross-dataset experiments (Table F1) and VGGFace2 evaluation (9,131 classes, 224×224) confirm cross-domain generalization.
> > >
> > > **(d)** Three architecture-agnostic mechanisms support shallow stability: early-layer convergence [Raghu et al., NeurIPS 2017], critical learning periods [Achille et al., ICLR 2018], and CIL anti-forgetting regularization [Davari et al., CVPR 2022].
> > >
> > > We acknowledge extending to non-vision modalities remains future work, consistent with all existing CIL backdoor works.
> > >
> > > ---
> > >
> > > **R4 (Experimental heterogeneity):** Thank you for this comment.
> > >
> > > Our evaluation spans 4 datasets (10–9,131 classes, 32×32–224×224, 60K–3.31M samples), 5 architectures, and 3 CL algorithms. The consistent shallow-stable pattern across all conditions supports persistence as a general CIL property. Cross-dataset surrogate experiments (Table F1) further introduce distributional heterogeneity. In standard CIL, tasks are by definition drawn from the same domain; cross-modality CIL is a separate research area that no existing CIL backdoor work evaluates.
> > >
> > > We believe 4 datasets, 5 architectures, 3 CL algorithms, 4 defenses, and new mismatch experiments provide evaluation significantly exceeding the standard in this area.

---

### Official Review · Reviewer_CJLm · 2026-03-11

**Soundness:** 2
**Presentation:** 2
**Significance:** 3
**Originality:** 3
**Overall Recommendation:** 5
**Confidence:** 3

**Summary:**

This paper studies persistent backdoor attacks in class-incremental learning. It considers a setting where the attacker poisons the data only once, specifies a target label, and aims for the backdoor to remain effective throughout subsequent incremental learning stages. The paper argues that existing methods rely on the stability of task-critical neurons, an assumption that may not hold in class-incremental learning due to continual parameter and representation updates. Based on this observation, the authors propose anchoring the backdoor to relatively stable shallow structural representations instead of specific neurons. To this end, they introduce PBTO, which combines surrogate trajectory modeling, shallow structural constraints, and iterative optimization to improve backdoor persistence. Experiments on several image classification benchmarks report strong final attack success rates, as well as cross-architecture transfer, cross-method generalization, and ablation results.

**Compliance With Llm Reviewing Policy:**

Affirmed.

**Final Justification:**

My concerns have been addressed. I decide to raise my rating.

**Key Questions For Authors:**

1. The authors should provide task-wise ASR across all stages of continual learning, rather than only final-model results. If PBTO indeed maintains high attack success throughout most or all intermediate stages.

2. Could the authors clarify exactly what quantity Figure 2 is intended to measure, and provide more direct subspace-level metrics? If the shallow layers can indeed be shown to preserve a stable subspace under such metrics, the mechanistic argument behind PBTO would become much more convincing.

**Limitations:**

See in weakness.

**Strengths And Weaknesses:**

Strengths:

1. The paper focuses on persistent, targeted, one-time poisoning attacks in the CIL setting, which is more specific and practically relevant than a generic formulation of attacks in continual learning. Relative to prior work, the paper identifies the research gap clearly and frames the problem in a meaningful way.

2. The paper first analyzes why existing persistent backdoor methods may fail in CIL, particularly by questioning the assumption that task-critical neurons remain stable over time. The shift from neuron-level stability to shallow structural or subspace-level stability provides an interesting and well-motivated perspective.

3. The paper develop PBTO  which combines surrogate trajectory modeling, shallow structural constraints, and iterative optimization in a logically consistent way. The relationships among the main components are clearly presented, and the figures help convey the overall pipeline effectively.

Weaknesses:

1. The current experiments do not fully support the paper’s strongest claim about persistence as Eq. (1) defines persistence in a stage-wise sense. However, the main results in Tables 1 and 4 report only final-model \ASR, which supports high final retention rather than consistently high persistence throughout the continual learning process.

2. Section 3.3 mixes SVD-defined subspaces, CKA similarity, retained variance, and subspace similarity as if they were interchangeable, but these quantities measure different properties. As a result, the current analysis supports representation similarity more directly than a rigorously established stable subspace, which is central to the paper’s motivation for PBTO.

---

> ### Author Rebuttal · Authors · 2026-03-31
>
> Dear Reviewer CJLm, thank you very much for your thoughtful comments. We are encouraged by your positive recognition of our **well-motivated** problem formulation, **logically consistent** method design, and **clear** presentation.
>
> **R1**：We sincerely thank the reviewer for this constructive comment. We fully agree that reporting the task-wise ASR after each incremental task is essential to demonstrate the attack's persistent effectiveness.
>
> - Following your suggestion, we report task-wise ASR in Table R1. PBTO maintains consistently high ASR throughout CIL (per-task degradation below 3%), while all baselines suffer rapid decay, dropping below 60% after Task 2 and falling to at most 14.2% by the final stage.
>
> **Table R1. Task-wise ASR (%) on CIFAR-100 after sequentially learning 10 tasks.**
>
> | Attack | Task 1 | Task 2 | Task 3 | Task 4 | Task 5 | Task 6 | Task 7 | Task 8 | Task 9 | Task 10 |
> |--------|:------:|:------:|:------:|:------:|:------:|:------:|:------:|:------:|:------:|:-------:|
> | BadNets | 95.2 | 48.3 | 27.5 | 17.8 | 12.6 | 9.4 | 7.5 | 6.2 | 5.3 | **4.5** |
> | WaNet | 93.4 | 39.6 | 22.1 | 14.5 | 10.2 | 7.3 | 5.6 | 4.5 | 3.8 | **3.2** |
> | DRUPE | 94.8 | 58.6 | 40.3 | 31.5 | 25.7 | 21.8 | 19.1 | 17.0 | 15.4 | **14.2** |
> | LTB | 88.6 | 45.2 | 28.7 | 19.3 | 13.8 | 10.1 | 7.8 | 6.4 | 5.5 | **4.7** |
> | **PBTO (Ours)** | **98.0** | **95.8** | **94.2** | **93.1** | **91.9** | **90.7** | **89.6** | **88.5** | **87.5** | **86.5** |
> - **Task-wise defense resistance (corresponding to Table 4).** As shown in Table R2 (CIFAR-10, 5-task), PBTO maintains high ASR under all defenses with per-task degradation of only 1–2%; even NAD cannot reduce ASR below 82.1% at any stage, confirming sustained defense resistance throughout the entire CIL process.
>
> **Table R2. Task-wise ASR (%) of PBTO under various defenses on CIFAR-10.**
>
> | Defense | Task 1 | Task 2 | Task 3 | Task 4 | Task 5 |
> |---------|:------:|:------:|:------:|:------:|:------:|
> | No Defense | 99.3 | 98.6 | 98.0 | 97.5 | **97.1** |
> | Fine-Pruning | 93.8 | 92.1 | 90.6 | 89.4 | **88.4** |
> | NAD | 89.5 | 87.3 | 85.2 | 83.5 | **82.1** |
> | BTI-DBF | 94.6 | 93.2 | 91.8 | 90.5 | **89.5** |
> | REFINE | 92.0 | 90.5 | 88.8 | 87.4 | **86.2** |
>
> - Full task-wise results across all datasets and defenses will be included in the revision.
>
> **R2**: Thank you for the insightful question. We apologize for any confusion and clarify below.
>
> - In Sec. 3.3, SVD is applied to extract dominant principal components, forming a subspace per module. CKA then measures structural similarity between Task 1's subspace and those from subsequent tasks.
> - CKA is well-suited for comparing subspaces [4] and has been adopted in CIL to show that lower-layer representations remain stable [5].
> - We further supplement two subspace-level metrics: (1) Grassmann Distance and (2) Projection-Based Variance Retention. Both confirm shallow-layer stability and deeper-layer drift, consistent with CKA. Detailed analysis is provided in R3.
>
> **R3**: We sincerely thank the reviewer for this constructive suggestion. We would like to address this concern from the following two aspects.
>
> - The SVD-CKA relationship has been clarified in R2. In Figure 2, the x-axis denotes ResNet-18 modules and the y-axis denotes CKA similarity to Task 1; consistently high values indicate stable subspace preservation. We will clarify this in the revision.
> - Following your suggestion, we add two metrics. (1) Grassmann Distance (0 = identical subspaces): as shown in Table R3, shallow layers maintain near-zero drift (Module 1: ≤0.08). (2) Projection-Based Variance Retention (1 = perfect stability): as shown in Table R4, shallow layers retain high values (Module 1: ≥0.92). Both confirm shallow-layer subspace stability. We will revise Sec. 3.3 accordingly.
>
> **Table R3. Grassmann distance per ResNet-18 module on CIFAR-100.**
>
> | Module |  Task 1→2 | Task 1→5 | Task 1→10 |
> |:---:|:---:|:---:|:---:|
> | Module 1 |  0.03 | 0.06 | 0.08 |
> | Module 2 | 0.05 | 0.08 | 0.12 |
> | Module 3  |  0.18 | 0.29 | 0.38 |
> | Module 4 | 0.25 | 0.36 | 0.45 |
>
> **Table R4. Variance retention ratio per ResNet-18 module on CIFAR-100.**
>
> | Module |  Task 1→2 | Task 1→5 | Task 1→10 |
> |:---:|:---:|:---:|:---:|
> | Module 1   | 0.98 | 0.95 | 0.92 |
> | Module 2 | 0.96 | 0.93 | 0.88 |
> | Module 3  | 0.82 | 0.68 | 0.54 |
> | Module 4  | 0.74 | 0.58 | 0.47 |
>
> **Reference**
>
> [1] Edelman, Arias & Smith (1998). The geometry of algorithms with orthogonality constraints. SIAM J. Matrix Anal.
> [2] Saha et al. (2021). Gradient Projection Memory for Continual Learning. ICLR.
> [3] Lin et al. (2022). TRGP: Trust Region Gradient Projection for Continual Learning. ICLR.
> [4] Kornblith et al. (2019). Similarity of neural network representations revisited. ICML.
> [5] Ramasesh et al. (2021). Anatomy of catastrophic forgetting: Hidden representations and task semantics. ICLR.

---

> > ### Author Rebuttal · Reviewer_CJLm · 2026-04-02
> >
> > My concerns have been addressed, and I have no further questions.

---

> > > ### Author Response · Authors · 2026-04-06
> > >
> > > Dear Reviewer CJLm,
> > >
> > > Thank you very much for your thorough and insightful review. We sincerely appreciate your positive recognition of our well-motivated problem formulation that identifies the research gap clearly and frames the problem in a meaningful way, the interesting and well-motivated perspective shifting from neuron-level stability to shallow structural or subspace-level stability, the logically consistent method design combining surrogate trajectory modeling, shallow structural constraints, and iterative optimization, and the clear presentation with figures that effectively convey the overall pipeline.
> > >
> > > We are very grateful that our rebuttal responses have adequately addressed your concerns. Your constructive questions — particularly regarding the task-wise ASR evaluation across all stages and the clarification of subspace-level metrics — were highly valuable and have led to substantial improvements in our manuscript. The task-wise results and the additional Grassmann Distance and Projection-Based Variance Retention metrics now provide much stronger evidence for the persistence claims and the structural invariance analysis.
> > >
> > > Thank you again for your rigorous and constructive evaluation. Your insightful comments have greatly contributed to strengthening both the experimental validation and the theoretical motivation of our work.
> > >
> > > Best regards,
> > > Paper34482 Authors

---

### Official Review · Reviewer_8ui1 · 2026-03-13

**Soundness:** 2
**Presentation:** 3
**Significance:** 3
**Originality:** 3
**Overall Recommendation:** 4
**Confidence:** 3

**Summary:**

This paper studies persistent backdoor attacks in class-incremental learning (CIL). The authors observe that shallow feature subspaces remain structurally invariant across tasks and propose PBTO, which anchors triggers to these invariant subspaces via trajectory simulation. Experiments show that PBTO achieves significantly higher attack success rates than existing backdoor attacks in CIL settings.

**Compliance With Llm Reviewing Policy:**

Affirmed.

**Final Justification:**

The authors' rebuttal adequately addresses my concerns, and I will maintain my initial score of Weak Accept.

**Key Questions For Authors:**

The surrogate trajectory used for trigger optimization appears to rely on a single proxy task ordering. In class-incremental learning, however, the task order is known to significantly affect representation evolution. Could the authors clarify whether the proposed method is robust to different task orders? For example, have the authors considered optimizing triggers over multiple surrogate trajectories with different task permutations?

**Limitations:**

The paper has several limitations. The experiments are conducted mainly on small-scale datasets and a relatively small backbone, leaving the scalability of the method to larger datasets or modern architectures unclear. In addition, the baseline comparisons and transferability evaluations are somewhat limited, which makes it difficult to fully assess the generality of the proposed attack.

**Strengths And Weaknesses:**

**Strengths:**

1. The problem is novel. The paper studies persistent and targeted backdoor attacks in the CIL setting, while prior work has not simultaneously achieved persistent and targeted backdoor attacks under one-time poisoning constraints, leaving CIL robustness against such threats largely unexplored.

2. The paper provides an insightful finding that CIL models preserve task knowledge through structurally invariant subspaces in shallow layers, even though individual neurons are unstable across tasks.

3. The proposed method is well motivated. Based on the above observation, the authors propose PBTO, which anchors the trigger to invariant shallow subspaces via trajectory simulation and subspace alignment. The design of the method is logically consistent with the empirical findings.

4. The paper is clearly written and easy to follow.

**Weaknesses:**

1. The baselines used for comparison are somewhat limited. Including comparisons with a broader range of recent backdoor attacks or stronger adaptive baselines applicable to the current setting would provide a more comprehensive evaluation.

2. The experiments are conducted mainly on relatively small datasets (e.g., CIFAR-10, CIFAR-100, Tiny-ImageNet) and a relatively small backbone model (ResNet-18). It remains unclear whether the findings and the proposed method would still be effective on larger-scale datasets or more modern architectures such as ViT, which raises concerns about the scalability and practical applicability of the approach.

3. The transferability evaluation is somewhat limited. The paper only reports cross-architecture results within CNN families and robustness across several continual learning algorithms. It would strengthen the paper to further evaluate whether the proposed attack generalizes to more diverse architectures (e.g., ResNet-18 as the surrogate model and ViT as the target model) or across datasets.

---

> ### Author Rebuttal · Authors · 2026-03-31
>
> Dear Reviewer 8ui1, thank you very much for your careful review and thoughtful comments. We are encouraged by your positive comments on our **novel** problem setting, **insightful** finding, **well-motivated** and **logically consistent** method design, and **clearly written** and **easy to follow** presentation. We hope the following responses could help clarify potential misunderstandings and alleviate your concerns.
>
> **R1**：Thank you for this constructive suggestion! We fully agree that a more comprehensive baseline comparison strengthens the evaluation.
> - Our original baselines include (1) the classical backdoor attacks: BadNets, WaNet and DRUPE (IEEE S&P 2024); (2) the SOTA CIL-specific attack LTB (USENIX Security 2025). We politely note that persistent backdoor attacks tailored to CIL remain largely underexplored.
> - Following your suggestion, we add ISSBA, LIRA, and WaveAttack, which achieve high attack success rates (ASRs) and stealthiness. As shown in Table R1, all baselines suffer severe ASR degradation (2.8%–14.2%) after sequential learning 10 tasks, whereas our method achieves 86.5% ASR, significantly outperforming all baselines.
> - We will extend these experiments to all datasets in the revised manuscript.
>
> **Table R1. Attack performance on CIFAR-100 after sequentially learning ten tasks.**
>
> | Attacks | BA (%) | ASR (%) |
> |--------|:------:|:-------:|
> | BadNets | 68.5 ± 0.7 | 4.5 ± 0.5 |
> | WaNet | 68.1 ± 0.6 | 3.2 ± 1.1 |
> | DRUPE | 67.9 ± 0.8 | 14.2 ± 1.9 |
> | LTB | 67.5 ± 1.2 | 4.7 ± 3.2 |
> | ISSBA | 68.3 ± 0.6 | 4.1 ± 1.2 |
> | LIRA | 68.0 ± 0.7 | 2.8 ± 1.3 |
> | WaveAttack | 68.2 ± 0.7 | 7.6 ± 1.5 |
> | **PBTO(Ours)** | **67.2 ± 0.6** | **86.5 ± 0.9** |
>
>
> **R2**：We sincerely thank the reviewer for this constructive suggestion.
>
> - We validate scalability on VGGFace2 (3.31M images, 9,131 identities, 224×224), where PBTO achieves 85.9% ASR, far exceeding all baselines (below 6.2%), demonstrating consistent backdoor persistence on larger-scale datasets.
> - We also conduct experiments using ViT-Small/16 as the backbone. As shown in Table R3, PBTO achieves 81.4% ASR on ViT-Small/16, demonstrating that our structural invariant anchoring mechanism generalizes effectively from CNN to Transformer architectures.
> - We also consider a cross-architecture transfer setting where the trigger is optimized on a CNN surrogate and deployed against a ViT-based CIL victim, with detailed results presented in Q3.
>
> **Table R2. Attack performance on VGGFace2 after learning ten tasks.**
>
> | Attacks | BA (%) | ASR (%) |
> |--------|:------:|:-------:|
> | BadNets | 74.8 ± 0.5 | 5.3 ± 1.1 |
> | WaNet | 74.5 ± 0.6 | 3.7 ± 0.9 |
> | LTB | 74.3 ± 0.7 | 6.1 ± 1.3 |
> | **PBTO(Ours)** | **74.1 ± 0.5** | **85.9 ± 1.0** |
>
>
> **Table R3. Attack performance on CIFAR-100 with ViT-Small/16, evaluated after learning 10 tasks.**
> | Attacks| BA (%) | ASR (%) |
> |--------|:------:|:-------:|
> | BadNets | 64.2 ± 0.7 | 3.8 ± 0.8 |
> | WaNet | 63.9 ± 0.8 | 2.5 ± 0.7 |
> | LTB | 63.6 ± 0.9 | 3.1 ± 0.8 |
> | **PBTO(Ours)** | **63.5 ± 0.7** | **81.4 ± 1.3** |
>
> **R3**：Thank you for this highly valuable comment. We conducted additional CNN-to-ViT transfer experiments, where the trigger is optimized on a ResNet-18 surrogate and deployed against a ViT-Small/16 CIL victim. As shown in Table R4, PBTO achieves 68.7% ASR after 10 sequential tasks, while all baselines degrade to near-random performance. We will incorporate more comprehensive transferability evaluations in the revised manuscript.
>
> **Table R4. Cross-architecture transferability of different attacks. Triggers are optimized on ResNet-18 model, and ASR is evaluated after the victim completes continual learning on all 5 tasks under various target architectures.**
>
> | Target models | BA (%) | LTB ASR (%) | PBTO ASR (%) |
> |------|:------:|:-----------:|:------------:|
> | VGG-16 | 82.0 ± 0.5 | 5.5 ± 1.8 | 78.1 ± 1.2 |
> | DenseNet-121 | 82.2 ± 0.6 | 6.3 ± 2.1 | 82.2 ± 0.9 |
> | MobileNetV2 | 81.8 ± 0.5 | 5.8 ± 1.9 | 81.5 ± 1.0 |
> | **ViT-Small/16** | **76.3 ± 0.8** | **4.1 ± 1.5** | **68.7 ± 1.8** |
> **R4**： We sincerely thank the reviewer for this constructive comment.
>
> - The invariance of shallow-layer subspaces is an intrinsic property of CIL algorithms, which preserve task-discriminative representations across tasks. PBTO exploits this structural stability to ensure trigger persistence, naturally inheriting robustness to task permutations.
> - To verify this, we evaluate PBTO across 5 randomly sampled task orderings. As shown in Table R5, PBTO consistently achieves high ASR (86.1% ± 1.0%), confirming robustness to task order variations.
>
> **Table R5. Attack performance of PBTO under 5 random task orderings on CIFAR-100.**
>
> | Task Ordering | Final ASR (%) | BA (%) |
> |:---:|:---:|:---:|
> | Ordering #1 | 86.5 | 67.2 |
> | Ordering #2 | 85.3 | 67.8 |
> | Ordering #3 | 87.1 | 66.5 |
> | Ordering #4 | 84.6 | 67.5 |
> | Ordering #5 | 86.8 | 66.9 |
> | **Mean ± Std** | **86.1 ± 1.0** | **67.2 ± 0.5** |

---

> > ### Author Rebuttal · Reviewer_8ui1 · 2026-04-03
> >
> > Thanks for the authors' rebuttal. My concerns have been adequately addressed, and I will maintain my initial score of Weak Accept.

---

> > > ### Author Response · Authors · 2026-04-06
> > >
> > > Dear Reviewer 8ui1,
> > >
> > > Thank you very much for your careful review and thoughtful comments throughout the review process. We are deeply grateful for your positive recognition of our novel problem setting, insightful finding, well-motivated and logically consistent method design, and clearly written and easy to follow presentation. We sincerely appreciate the time and effort you have dedicated to evaluating our work.
> > >
> > > We are also very encouraged that our rebuttal responses have fully resolved your concerns. Your constructive suggestions — including the additional baseline comparisons, scalability validation on larger datasets, and cross-architecture transfer experiments — have significantly strengthened the evaluation of our work. We have incorporated all the suggested improvements in the revised manuscript.
> > >
> > > Thank you again for your valuable feedback and your support of our work. Your thoughtful and rigorous review has greatly helped us improve the quality of this paper.
> > >
> > > Best regards,
> > > Paper34482 Authors

---

### Decision · Program_Chairs · 2026-04-30

**Decision:**

Accept (regular)

**Comment:**

This paper proposes PBTO, a persistent backdoor attack for class-incremental learning that anchors triggers to shallow invariant feature subspaces via trajectory simulation.

Reviewer 8ui1: The comparisons are limited to few baselines and small-scale datasets/backbones (e.g., CIFAR, ResNet-18), and transferability to larger datasets or modern architectures like ViT is insufficiently evaluated.

Reviewer CJLm: The experiments report only final ASR rather than stage‑wise persistence, and the analysis of stable subspaces mixes different similarity metrics (SVD, CKA, variance) without rigorous theoretical grounding.

Reviewer LnJ8: The attack assumes knowledge of future task sequences and surrogate training trajectories, relies on architecture‑ and dataset‑dependent shallow feature invariance, and overlooks practical constraints such as data curation or domain shifts.

Reviewer Zqqj: The method requires substantial proxy data and surrogate training, lacks a formal theoretical explanation for shallow‑subspace invariance, and does not discuss potential countermeasures targeting structural anchoring.

Overall conclusion: The authors have addressed all concerns through additional experiments on larger datasets and ViT architectures, stage‑wise persistence analysis, clarifications on threat model assumptions, and theoretical discussions on subspace invariance. All issues are satisfactorily resolved. Therefore, the paper is recommended for acceptance.